# $f$-Trajectory Balance: A Loss Family for Tuning GFlowNets, Generative Models, and LLMs with Off- and On-Policy Data

**Jake Fawkes** [1]   **Jason Hartford** [2,3]

## Abstract

In GFlowNets and variational inference, it has been shown that the mean square error between target and model log probabilities is an effective, low variance, surrogate loss for training generative models. This loss has the property that when evaluated *on-policy* its gradients correspond to those of the KL divergence, while *off-policy* it remains a valid loss with the same global minimizer. In this work, we demonstrate that this construction can be extended to the whole family of $f$-divergences, leading to a family of losses whose on-policy gradients are that of the corresponding $f$-divergence, but retain the same global minimizer off-policy. Specifically, we show that the on-policy gradients lead to a one-to-one correspondence between translation invariant loss functions on the target and model log probabilities, and $f$-divergences. This equivalence allows us to design new surrogate loss functions for tuning a wide class of generative models that inherit the properties of the corresponding $f$-divergence, such as being more mode covering, whilst being applicable to off-policy data. We apply our losses on a range of tasks, including classic synthetic examples, SynFlowNets for molecule discovery, and asynchronous large language model (LLM) tuning, demonstrating that our models retain their predicted properties on- and off-policy in a wide class of generative models.

## 1. Introduction

The alignment and fine-tuning of generative models—whether they are large language models (LLMs) generating text, or Generative Flow Networks (GFlowNets) generating molecular graphs—relies on minimizing the discrepancy between the model's distribution. In Reinforcement Learning (RL) fine-tuning, this objective is often framed as maximizing expected reward subject to a KL-divergence penalty, or equivalently, minimizing a divergence between the policy and the target distribution.

Traditionally, optimizing these objectives involves gradient estimators with high variance, such as REINFORCE (Williams, 1992), or sophisticated policy gradient methods (e.g. Schulman et al., 2017). However, recently the GFlowNet literature and large scale LLM tuning in Kimi (Team et al., 2025a) has used a simpler approach: minimizing the mean squared error (MSE) between the log-probabilities of the model and the target. This has similarities with the K2 estimator of the $\mathbb{KL}$ divergence proposed in Schulman (2020), which we term $\mathbb{KL}_{sq}$. Tang and Munos (2025) note this "squared KL" loss has a surprising property: whilst it is a biased estimator of the $\mathbb{KL}$, its gradients on on-policy data are unbiased to the true KL gradients. Further, as it is built on the MSE, it remains a valid loss function with the same global minimizer when applied to off-policy data (Bartoldson et al., 2025; Tang et al., 2025).

While the $\mathbb{KL}_{sq}$ estimator offers stability and off-policy compatibility, by construction, it inherits the properties of the reverse KL divergence. In particular, models trained under this objective will typically exhibit "mode-seeking" behavior where the model collapses onto relatively few high-probability modes rather than covering the full diversity of the target distribution. In many generative tasks, such as drug discovery or exploratory agents, "mode-covering" behavior is often preferred because generative models are used to explore the space of high reward candidates (e.g. all possible drug targets), rather than to seek out one (or a few) particularly good candidate(s).

In this work, we demonstrate that the effectiveness of the $\mathbb{KL}_{sq}$ loss is not a unique phenomenon restricted to the KL divergence by deriving analogous losses for the entire family of $f$-divergences. We establish a theoretical equivalence showing that *any* translation-invariant loss function on log-probabilities corresponds to a specific $f$-divergence. This allows us to derive a new family of surrogate losses that

---

[1]Department of Statistics, University College London, UK *(work done while an intern at Valence Labs)*. [2]Valence Labs, London, UK [3]Recursion, London, UK. Correspondence to: Jake Fawkes <jakefawkess@gmail.com>, Jason Hartford <jason@valencelabs.com>.

*Proceedings of the $43^{rd}$ International Conference on Machine Learning*, Seoul, South Korea. PMLR 306, 2026. Copyright 2026 by the author(s).

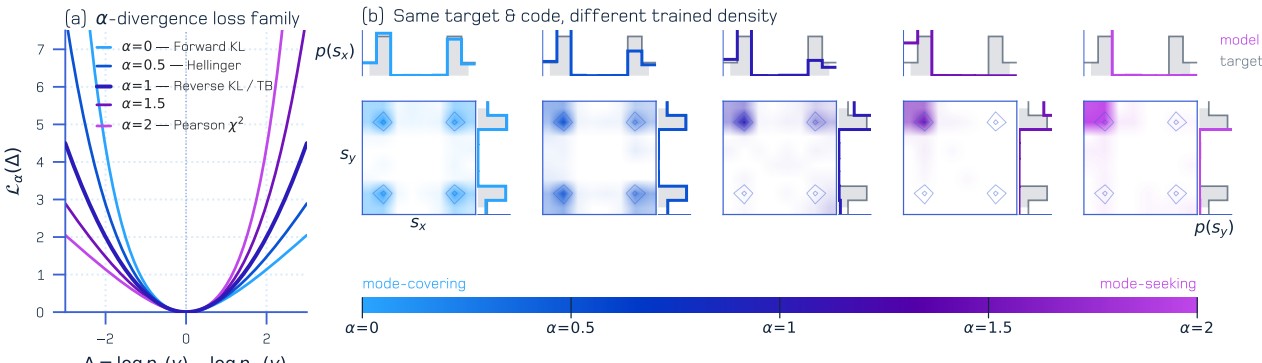

*Figure 1.* $f$-divergences include the $\alpha$-divergence family which contains both Forward KL ($\alpha=0$) and Reverse KL / standard Trajectory Balance ($\alpha=1$) as special cases. Lower $\alpha$ yields a more mode-covering loss, higher $\alpha$ a more mode-seeking one.

inherit the specific properties of their parent divergences (e.g., the mode-covering behavior of the Forward KL or Hellinger distance) while retaining the optimization benefits of the $\mathbb{KL}_{sq}$ estimator: low-variance gradients on-policy and validity off-policy.

We demonstrate the behaviour and utility of this family of loss functions across a variety of tasks. On the Hypergrid task (Bengio et al., 2021)—a synthetic task explicitly designed to test mode coverage—we show that the Hellinger and forward KL losses discover more modes of the reverse KL loss implied by the trajectory balance loss (Malkin et al., 2022b). We then apply these losses to both molecule generation—by modifying the losses in SynFlowNet (Cretu et al., 2025)—, class conditional sampling in diffusion models and asynchronous reinforcement learning on language models. In both cases, we found that we were able to learn higher entropy policies with mode-covering instances of our family of surrogate loss functions. In summary, our contributions are as follows:

- We derive a general family of translation-invariant surrogate loss functions, $\mathcal{L}_f$, whose expected auto-differentiated gradients match those of the corresponding $f$-divergence on-policy.

- We prove the inverse relationship: any convex, translation-invariant loss on log-probabilities corresponds to minimizing a specific $f$-divergence.

- We generalize the "Vargrad" (Richter et al., 2020) and batch-wise normalization techniques to arbitrary $f$-divergences, addressing the intractable partition function problem in RL fine-tuning.

- We apply this framework to GFlowNets, showing that our losses generalize the standard Trajectory Balance objective (Malkin et al., 2022a), and empirically demonstrate on synthetic grids and SynFlowNet

molecular discovery tasks that we can control the exploration-exploitation trade-off (mode-seeking vs. mode-covering) simply by changing the surrogate loss function.

## 2. Related work

**GFlowNets and GFlowNet losses**  Bengio et al. (2021) introduce GFlowNets and train them via the flow matching loss, which applies directly to a neural network aiming to learn the optimal Markovian flow for a given GFlowNets and applies on an action level. The detailed balance objective was proposed in Bengio et al. (2023), which again applies to the action level but parametrises a forward and backward policy distribution. Malkin et al. (2022a) propose trajectory balance, which enforces an equality constraint between the forward policy, backward policy, and normalisation constant on the trajectory level and penalises it with the mean square error. It was shown in Malkin et al. (2023) that on-policy training with this loss is equivalent to a form of hierarchical variational inference (Ranganath et al., 2016) with the $\mathbb{KL}$ divergence. Our results extend these to all $f$-divergences.

The Vargrad objective (Richter et al., 2020) is often used in GFlowNet training, which use the same square error form as Malkin et al. (2022a) but estimate the normalisation constant in a batch-wise fashion. This Vargrad loss comes from the log variance divergence introduced in Nüsken and Richter (2020), where they also propose a variance based estimator built of the Pearson $\chi^2$ divergence which is most similar to our work. However, this would not share the same property of matching on-policy gradients as we show in the paper that the correct way to extend this property is using generalised deviations (Rockafellar et al., 2006) instead of the variance. Silva et al. (2024) discuss using alternative divergence measures for training GFlowNets, however their work focuses on the on-policy setting. Most related to our work is Hu et al. (2025) who derive similar integral conditions to generalise

the trajectory balance and KL relationship to the family of $f$-divergences. However, their work does not provide the formal proof that their mapping produces a loss function, which is required for off policy training. They also do not include a generalisation of the VarGrad loss.

**$f$-Divergences and Generative Models** $f$-divergences (Ali and Silvey, 1966; Morimoto, 1963) have an extensive history in the tuning of probabilistic models (Minka et al., 2005). In the context of generative models, they have been applied heavily in variational inference (Wang et al., 2018) as well as to GANs (Nowozin et al., 2016), and more recently some work on applies $f$-divergences to tuning diffusion models (Novello et al., 2025; Tang, 2024). $f$-divergences have been applied in the context of LLMS, either directly as an objective (Go et al., 2023; Han et al., 2024) or as a regularised (Huang et al.). However to the best of our knowledge, none of this work demonstrates off policy validity, an important aspect as much LLM tuning is done asynchronously (Intellect, 2025) or completely off-policy. $f$-divergences have also been applied extensively to imitation learning (Ghasemipour et al., 2020; Ke et al., 2020) and in RL for policy gradients (Agarwal et al., 2023).

## 3. Motivation In RL Tuning LLMs

**RL tuning LLMs and KL gradients** We use $\pi_\theta$ to denote the LLM which is viewed as a policy over sequences of tokens, $\mathbf{y}$, given a prompt $\mathbf{x}$. Reinforcement learning is then used to tune this model by maximizing the following reward for a distribution $P(\mathbf{x})$ over prompts/states $\mathbf{x}$:

$$\mathcal{J}(\theta) = \mathbb{E}_{\mathbf{x} \sim P(\mathbf{x})} \left[ \mathbb{E}_{\mathbf{y} \sim \pi_\theta(\mathbf{y}|\mathbf{x})} [r(\mathbf{x}, \mathbf{y})] \right], \quad (1)$$
$$- \beta \mathbb{KL} \left( \pi_\theta \left( \cdot | \mathbf{x} \right), \pi_{\text{ref}} \left( \cdot | \mathbf{x} \right) \right). \quad (2)$$

This objective is then generally optimised via PPO (Schulman et al., 2017) or REINFORCE (Williams, 1992). However, the gradient of the $\mathbb{KL}$ term cannot be obtained via auto-differentiation through a Monte Carlo estimator of the $\mathbb{KL}$, as Tang and Munos (2025) noted many open-source implementations did at the time. This is because the $\mathbb{KL}$ depends on $\theta$ both in terms of its sampling and its objective, where auto-differentiation only accounts for the latter. The correct gradient is:

$$\nabla_\theta \mathbb{KL} = \mathbb{E}_{\pi_\theta(\mathbf{y}|\mathbf{x})} \left[ \log \frac{\pi_\theta(\mathbf{y}|\mathbf{x})}{\pi(\mathbf{y}|\mathbf{x})} \nabla_\theta \log_\theta \pi_\theta(\mathbf{y}|\mathbf{x}) \right],$$

whereas a naive auto-differentiation would give the gradient as:

$$\mathbb{E}_{\pi_\theta} \left[ \nabla_\theta \log \frac{\pi_\theta(\mathbf{y}|\mathbf{x})}{\pi(\mathbf{y}|\mathbf{x})} \right] = \mathbb{E}_{\pi_\theta} [\nabla_\theta \log \pi_\theta(\mathbf{y}|\mathbf{x})],$$

where this expression's expectation is zero. Tang and Munos (2025) suggest using the standard REINFORCE estimator

of the gradient, but also discuss the squared $\mathbb{KL}$ estimator of Schulman (2020). This is denoted $\mathbb{KL}_{\text{sq}}$ and given by:

$$\mathbb{KL}_{\text{sq}}(\pi_\theta \| \pi_{\text{ref}})(\mathbf{x}) = \frac{1}{2} \mathbb{E}_{\mathbf{y} \sim \pi_\theta} \left[ \left( \log \frac{\pi_\theta(\mathbf{y}|\mathbf{x})}{\pi_{\text{ref}}(\mathbf{y}|\mathbf{x})} \right)^2 \right],$$

where we have written this as a function of the conditioning value $\mathbf{x}$. This is itself biased estimator of the $\mathbb{KL}$, however it has the property that the naive auto-differentiated gradient is in expectation the gradient of the true $\mathbb{KL}$. Specifically,

$$\mathbb{E}_{\mathbf{y} \sim \pi_\theta} \left[ \nabla_\theta \frac{1}{2} \left( \log \frac{\pi_\theta(\mathbf{y}|\mathbf{x})}{\pi_{\text{ref}}(\mathbf{y}|\mathbf{x})} \right)^2 \right] = \nabla_\theta \mathbb{KL}(\pi_\theta \| \pi_{\text{ref}})(\mathbf{x}).$$

**A single loss for off- and on-policy RL from the $\mathbb{KL}$ divergence** We now describe how the $\mathbb{KL}_{sq}$ estimator leads to a loss that can be used to optimise the objective in Equation 1 using either off or on-policy data, as in Tang et al. (2025) and Bartoldson et al. (2025). First, we have that $\mathcal{J}(\theta)$ may be written as:

$$\mathcal{J}(\theta) = \beta \mathbb{E}_{\mathbf{x} \sim P(\mathbf{x})} \left[ \mathbb{E}_{\mathbf{y} \sim \pi_\theta(\mathbf{y}|\mathbf{x})} \left[ \log e^{\beta^{-1} r(\mathbf{x}, \mathbf{y})} \right] \right],$$
$$- \beta \mathbb{E}_{\mathbf{x} \sim P(\mathbf{x})} \left[ \mathbb{E}_{\mathbf{y} \sim \pi_\theta(\mathbf{y}|\mathbf{x})} \left[ \log \frac{\pi_\theta(\mathbf{y}|\mathbf{x})}{\pi_{\text{ref}}(\mathbf{y}|\mathbf{x})} \right] \right]$$
$$= \beta \mathbb{E}_{x \sim P(\mathbf{x})} \left[ -\mathbb{KL} \left( \pi_\theta \| \pi_\star \right) (\mathbf{x}) \right] + C$$

where $C$ is a constant independent of $\theta$ and $\pi_\star$ defined as:

$$\pi_\star(\mathbf{y}|\mathbf{x}) = \frac{\pi_{\text{ref}}(\mathbf{y} \mid \mathbf{x}) \exp \left( \beta^{-1} r(\mathbf{x}, \mathbf{y}) \right)}{Z(\mathbf{x})},$$

$Z(\mathbf{x}) = \int \pi_{\text{ref}}(\mathbf{y} \mid \mathbf{x}) \exp \left( \beta^{-1} r(\mathbf{x}, \mathbf{y}) \right) d\mathbf{y}$ is the partition function. Therefore maximisation of equation 1 corresponds to minimising $\mathbb{E}_{x \sim P(\mathbf{x})} \left[ -\mathbb{KL} \left( \pi_\theta \| \pi_\star \right) (\mathbf{x}) \right]$. If we assume for now that we know the partition function, $Z(\mathbf{x})$ then we can make use of the $\mathbb{KL}_{sq}$ property to define the loss $\mathcal{L}_{\mathbb{KL}_{sq}}(\mathbf{x}, \mathbf{y}, \theta)$ as,

$$\mathcal{L}_{\mathbb{KL}_{sq}}(\mathbf{x}, \mathbf{y}, \theta) := \frac{1}{2} \left( \log \frac{\pi_\theta(\mathbf{y}|\mathbf{x})}{\pi_\star(\mathbf{y}|\mathbf{x})} \right)^2$$

where this loss has the property that $\mathbb{E}_{\mathbf{y} \sim \pi_\theta(\mathbf{y}|\mathbf{x})} [\nabla \mathcal{L}_{\mathbb{KL}}(\mathbf{x}, \mathbf{y}, \theta)] = \nabla_\theta \mathbb{KL}(\pi_\theta \| \pi_{\text{ref}})(\mathbf{x})$, so that the on-policy gradients are equal in expectation to the gradients of the $\mathbb{KL}$. This allows us to define an objective:

$$\tilde{\mathcal{J}}(\theta) = \mathbb{E}_{\mathbf{x} \sim P(\mathbf{x})} \left[ \mathbb{E}_{\mathbf{y} \sim \pi_\theta(\mathbf{y}|\mathbf{x})} \left[ \mathcal{L}_{\mathbb{KL}_{sq}}(\mathbf{x}, \mathbf{y}, \theta) \right] \right],$$

where the expected auto-differentiated gradients of $\tilde{\mathcal{J}}(\theta)$ are equal to the REINFORCE gradients of $\mathcal{J}(\theta)$. Seeing as $\mathcal{L}_{\mathbb{KL}_{sq}}$ is simply the squared loss between the logprobs, this has the additional benefit that if we instead sampled the completions off-policy from some distribution $\mu$ and set the objective as:

$$\tilde{\mathcal{J}}_\mu(\theta) = \mathbb{E}_{\mathbf{x} \sim P(\mathbf{x})} \left[ \mathbb{E}_{\mathbf{y} \sim \mu(\mathbf{y}|\mathbf{x})} \left[ \mathcal{L}_{\mathbb{KL}_{sq}}(\mathbf{x}, \mathbf{y}, \theta) \right] \right],$$

then if the support of $\pi_\theta(\cdot|\mathbf{x})$ is contained in $\mu(\cdot|\mathbf{x})$ for each $\mathbf{x}$ these share the same minimizer. This follows from the fact that $\tilde{\mathcal{J}}_\mu(\theta)$ is minimised by minimising $\mathbb{E}_{\mathbf{y}\sim\mu(\mathbf{y}|\mathbf{x})}\left[\mathcal{L}_{\mathbb{KL}_{sq}}(\mathbf{x},\mathbf{y},\theta)\right]$ for each $\mathbf{x}$ which occurs exactly when $\pi_\theta(\mathbf{y}|\mathbf{x})=\pi_\star(\mathbf{y}|\mathbf{x})$ for all $\mathbf{y}$ in the support of $\mu(\cdot|\mathbf{x})$.

**Unnormalised Target Distribution**   If $Z(\mathbf{x})$ is not known, which is commonly the case, we can use a batch based estimate for the normalisation constant, which recovers the VarGrad estimator (Richter et al., 2020) of the $\mathbb{KL}$ gradient. That is, for a sampled batch of completions $\mathcal{B}=\{\mathbf{y}_1,\ldots,\mathbf{y}_B\}$ for each $\mathbf{x}$ we can estimate $\log Z(\mathbf{x})$ as:

$$\widehat{\log Z(\mathbf{x})} = \frac{1}{B}\sum_{i=1}^B\left(\frac{r(\mathbf{y}_i,\mathbf{x})}{\beta}+\log\frac{\pi_{\text{ref}}(\mathbf{y}_i|\mathbf{x})}{\pi_\theta(\mathbf{y}_i|\mathbf{x})}\right) \quad (3)$$

to form the batch wise Vargrad loss as:

$$\mathcal{L}_{\text{KL}}^{\text{VG}}(\mathcal{B},\theta) = \frac{1}{B}\sum_{i=1}^B\left[\log\frac{\text{SG}\left[\widehat{Z}(\mathbf{x})\right]\pi_\theta(\mathbf{y}_i\mid\mathbf{x})}{\pi_{\text{ref}}(\mathbf{y}_i\mid\mathbf{x})e^{r(\mathbf{x},\mathbf{y}_i)/\beta}}\right]^2$$

where SG $[\cdot]$ is the stop gradient function.

A version of this loss was applied in Kimi K2 (Team et al., 2025a) and K1.5 (Team et al., 2025b), where the generating policy, $\theta_{\text{old}}$, is used as the reference distribution and everything is scaled by $\beta$. This gives the following loss:

$$\mathcal{L}^{\text{K2}}(\mathcal{B},\theta) = \frac{1}{B}\sum_{i=1}^B\left[\beta\log\frac{Z\pi_\theta(\mathbf{y}_i\mid\mathbf{x})}{\pi_{\theta_{\text{old}}}(\mathbf{y}_i\mid\mathbf{x})}-r(\mathbf{y}_i,\mathbf{x})\right]^2.$$

These papers use the mean rewards, $\bar{r}(\mathbf{x})=\frac{1}{B}\sum_{i=1}^B r(\mathbf{x},\mathbf{y}_i)$, as an estimate for $\log Z$ instead of equation 3. This can be viewed as assuming that $0\approx\beta\left(\log\pi_\theta(\mathbf{y}_i|\mathbf{x})-\log\pi_{\theta_{\text{old}}}(\mathbf{y}_i|\mathbf{x})\right)$, which will hold exactly if the baseline is on-policy, i.e $\theta=\theta_{\text{old}}$, and approximately if the baseline is slightly off-policy or $\beta$ is small. This has the advantage of not needing a forward pass for the whole batch to calculate $\widehat{\log Z}$.

## 4. Generalising Vargrad to Arbitrary Deviations and *f*- divergences

In this section we show that the approaches of the previous section can be generalised to arbitrary *f*-divergences. Specifically we can start from any *f*-divergence and construct a surrogate loss $\mathcal{L}_f$ whose gradients match the *f*-divergence on-policy and remains a valid loss with the same minimizer off-policy. Moreover, we show the reverse implication, that using any translation invariant loss on the logprobs of the current and target distribution corresponds to minimising a corresponding *f*-divergence, and that these

maps are the inverse of each other. First we give the definition of an *f*-divergence. Throughout we drop the conditioning on $\mathbf{x}$ for clarity.

**Definition 4.1.** Let $f$ be a convex function with $f(1)=0$. The associated *f*- divergence is then defined as:

$$\mathcal{D}_f(p\|q) = \mathbb{E}_{\mathbf{y}\sim q(\mathbf{y})}\left[f\left(\frac{p(\mathbf{y})}{q(\mathbf{y})}\right)\right].$$

We add the additional requirements that $f'(1)=1$ and $f''(1)=1$ so that the scale of all $f$ divergences is standardised. This can always be achieved as $\tilde{f}(x)=\lambda_1 f(x)+\lambda_2(x-1)$ is convex with $f(1)=0$ for $\lambda_i\geq 0$ and that $\mathcal{D}_{\tilde{f}}(p\|q)=\lambda_1\mathcal{D}_f(p\|q)$ so that the scaling does not change the shape of the divergence.

*f*-divergences are not symmetric and throughout we will write the target first as $\mathcal{D}_f(p_\theta\|p_\star)$. As for any convex function, $\tilde{f}(t)=\frac{1}{t}\tilde{f}\left(\frac{1}{t}\right)$ is convex, the reverse divergence, $\mathcal{D}_f(p_\theta\|p^*)$ is equal to $\mathcal{D}_{\tilde{f}}(p^*\|p_\theta)$ and so can always be written in this form. Given this, the $\mathbb{KL}$ discussed in Section 3 is the from now on the reverse $\mathbb{KL}$ and corresponds to choosing $f(t)=t\log t$. The gradient of an *f*-divergence is given by the log derivative trick as:

$$\nabla_\theta D_f(p_\theta\|p_\star) = \mathbb{E}_{\mathbf{y}\sim p_\theta}\left[f'\left(\frac{p_\theta(\mathbf{y})}{p_\star(\mathbf{y})}\right)\nabla_\theta\log p_\theta(\mathbf{y})\right].$$

Now given this we define the following loss function, which generalises the $\mathbb{KL}_{\text{sq}}$ loss to arbitrary *f*-divergences in the sense that the gradients are equivalent on-policy but it is also a valid off-policy loss:

**Proposition 4.2.** *Let* $\Delta_\theta(\mathbf{y}) = \log p_\theta(\mathbf{y})-\log p_\star(\mathbf{y})$. *Then the function* $\mathcal{L}_f : \mathbb{R}\to\mathbb{R}$ *given by:*

$$\boxed{\mathcal{L}_f(\Delta_\theta(\mathbf{y})) = \int_0^{\Delta_\theta(\mathbf{y})} f'(\exp(t)) - f'(1)dt}$$

*is a translation invariant loss function that generalises the construction in Section 3 in that:*

1. *The population loss between the log probabilities of the target and current distribution:*

$$\mathcal{J}_{\mu,f}(\theta) = \mathbb{E}_{\mathbf{y}\sim\mu}\left[\mathcal{L}_f\left(\Delta_\theta(\mathbf{y})\right)\right],$$

*is minimised when* $p_\theta = p_\star$ *so long as the support of* $p_\star$ *is contained in the support of* $\mu$.

2. *If the samples are generated on-policy* ($\mu = p_\theta$), *the expected auto-differentiated gradients of* $\mathcal{J}_{\mu,f}(\theta)$ *correspond to the gradients of the associated f-divergence:*

$$\nabla_\theta\mathcal{D}_f(p_\theta\|p_\star) = \mathbb{E}_{\mathbf{y}\sim p_\theta}\left[\nabla_\theta\mathcal{L}_f\left(\Delta_\theta(\mathbf{y})\right)\right]$$

Applying this construction to the reverse $\mathbb{KL}$ divergence directly recovers the loss in Section 3:

**Example 4.3.** For the reverse $\mathbb{KL}$ divergence we have $f(u) = u \log(u)$. This gives $f'(u) = \log(u) + 1$, $f'(1) = 1$, and:

$$\mathcal{L}_{u \log(u)}(\Delta_\theta(\mathbf{y})) = \int_0^{\Delta_\theta(\mathbf{y})} \log(\exp(t)) dt$$
$$= \frac{1}{2} \left(\Delta_\theta(\mathbf{y})\right)^2$$

In fact, this result is not specific to the square error loss and works for any *translation invariant loss function*. Specifically let $\ell : \mathbb{R} \to \mathbb{R}$ be some strictly convex, differentiable function that is minimized at $\ell(0)$, with $\ell(y - \hat{y})$ giving the loss between the prediction $\hat{\mathbf{y}}$ and the target $\mathbf{y}$. For all such loss functions, we can show the reverse result that using $\ell(\cdot)$ on the difference between target and model logprobs can be associated with a particular $f$ divergence using its on-policy gradients. Moreover, these maps are inverses of each other:

**Proposition 4.4.** *Let $\ell : \mathbb{R} \to \mathbb{R}$ be a translation invariant loss function. Then the objective:*

$$\tilde{\mathcal{J}}_{\mu,\ell}(\theta) = \mathbb{E}_{\mathbf{y} \sim \mu} \left[ \ell\left(\Delta_\theta(\mathbf{y})\right) \right],$$

*is minimised when $p_\theta = p_\star$ so long as the support of $p_\star$ is contained in the support of $\mu$. Further it has the property that the its auto-differentiated gradients correspond to the gradients of a corresponding $f$-divergence:*

$$\mathbb{E}_{\mathbf{x} \sim p_\theta(\mathbf{y})} \left[ \nabla_\theta \ell(\Delta_\theta(\mathbf{y})) \right] = \nabla_\theta \mathcal{D}_{f_\ell} \left( p_\theta \| p_\star \right),$$

*where $f_\ell$ is defined as:*

$$f_\ell(u) = \lambda_1 \int_1^u \ell'(\log t) dt + \lambda_2(u - 1) + c,$$

*for $\lambda_i, c \in \mathbb{R}$ chosen to satisfy $f_\ell(1) = 0$, $f_\ell'(1) = 1$, and $f_\ell''(1) = 1$. Moreover this mapping is the inverse of that in Proposition 4.2 in the sense that $f_{\mathcal{L}_f} = f$ and $\mathcal{L}_{f_\ell} = \ell$.*

The boundary conditions here ensure that the maps are the inverse of each other, but the gradient equivalence for all $\lambda_i, c$. However, this choice of parameter values does ensure that the gradient is equivalently scaled across divergences as the model approaches convergence. Appendix A.2.1, shows that starting from the squared loss we return to the $\mathbb{KL}$.

### 4.1. DevGrad Loss for Batch-Wise Normalisation

We now introduce the generalisation of the Vargrad loss, which can be applied to unnormalised target distributions. We refer to this as the *DevGrad* loss, as it corresponds to replacing the batch-wise variance with the batch-wise *generalised deviation* (Rockafellar and Uryasev, 2013; Rockafellar et al., 2006) of the difference in logprobs. This also

has the property of centring the score function coefficients in the batch, which leads to reduced variance even if we are working with a normalised target distribution.

Let the target density be of the form $p_\star(\mathbf{y}) = \frac{1}{Z} \exp(\mathcal{R}(\mathbf{y}))$ where $\mathcal{R}(\mathbf{y})$ is the reward function or (negative) energy function and $Z = \int \exp(\mathcal{R}(\mathbf{y})) d\mathbf{y}$ is the partition function. The partition function remains intractable, but we can use the same approach as Vargrad and estimate it as:

$$\widehat{\log Z} = \min_C \frac{1}{B} \sum_i \mathcal{L}_f \left( \Delta(\mathbf{y}_i) + C \right). \tag{4}$$

where $\Delta(\mathbf{y}_i) = \log p_\theta(\mathbf{y}_i) - \mathcal{R}(\mathbf{y}_i)$ are the unnormalised difference in logprobs/energy functions. Given this we define the *Devgrad* loss as follows:

**Definition 4.5.** Given a loss function, $\mathcal{L}_f$, defined as in Section 4, we define the batch-wise **Devgrad** loss for a batch $\mathcal{B} = \{\mathbf{y}_1, \ldots, \mathbf{y}_B\}$ as:

$$\mathcal{L}_f^{\mathrm{DG}}(\mathcal{B}, \theta) = \frac{1}{B} \sum_i \mathcal{L}_f \left( \Delta(\mathbf{y}_i) + \mathrm{SG} \left[ \widehat{\log Z} \right] \right).$$

where $\widehat{\log Z}$ satisfies Equation 4.

To see this is a valid loss, note that if we have reached the optimum, the difference between the target and current log-probs for the generation will be independent of $\mathbf{y}$ and so have deviation zero for any generalised deviation measure. Moreover, we have that if the generalised deviation measure is zero for every batch the target logprobs equal the energy function plus a constant, and so taking the exponential the learned distribution is proportional to the target distribution and therefore equal to it. This demonstrates that we have converged to the correct distribution if and only if the generalised deviation measure is zero for all batches in the support of the sample.

Again, this corresponds to taking the batch wise *generalised deviation* (Rockafellar and Uryasev, 2013; Rockafellar et al., 2006) of the batch wise difference in logprobs. When $\mathcal{L}_f(y) = y^2$ this recovers the variance, but for other $\mathcal{L}_f$ we recover different deviations, such as the mean absolute deviation around the median when $\phi_f(y) = |y|$, which in terms of $f$-divergences corresponds to the total variation. As we show in Appendix B, using $\widehat{\log Z}$ also centres the batch wise score function coefficients, which leads to variance reduction in the same way as Vargrad.

### 4.2. Tempered Loss

If the posterior probabilities are defined by the reward tilted distribution as:

$$\pi_\star(\mathbf{y}|\mathbf{x}) = \frac{1}{Z(\mathbf{x})} \pi_{\mathrm{ref}}(\mathbf{y} \mid \mathbf{x}) \exp \left( \beta^{-1} r(\mathbf{x}, \mathbf{y}) \right), \tag{5}$$

for small $\beta$ we can run into problems of exploding loss and therefore gradients due to the $\exp\left(\beta^{-1}r(\mathbf{x},\mathbf{y})\right)$ term. For example, if the reward is bounded between $[0,1]$ and $\beta = 0.005$ the unnormalised probabilities can be of magnitude $e^{200}$, causing significant numerical overflow. Alternatively, if we were to use clipping we would end up losing a significant amount of signal. E.g. clipping the exponential at values greater than $e^{20}$ would leave us unable to differentiate between samples with rewards greater than 0.1.

To resolve this, we first define the tempered distribution:

**Definition 4.6.** For a distribution $p(\mathbf{y}|\mathbf{x})$, we define the tempered distribution $\tilde{p}_\beta(\mathbf{y}|\mathbf{x})$ as the distribution that is proportional to $p^\beta(\mathbf{y}|\mathbf{x})$. In terms of energy functions we have that if $p(\mathbf{y}|\mathbf{x}) = \frac{1}{Z(\mathbf{x})}\exp(\mathcal{R}(\mathbf{y}|\mathbf{x}))$ then:

$$p^\beta(\mathbf{y}|\mathbf{x}) = \frac{1}{\tilde{Z}(\mathbf{x})}\exp(\beta\mathcal{R}(\mathbf{y}|\mathbf{x})). \qquad (6)$$

Now, as mentioned above, for the tilted distribution $\pi_\star$ given in Equation 5, we have the problem that the magnitude of the energy function $\mathcal{R}_\star(\mathbf{y}|\mathbf{x})$ scales with $\frac{1}{\beta}$ making training with small $\beta$ impractical. However, the energy function of the tempered distribution is $\beta\mathcal{R}_\star(\mathbf{y}|\mathbf{x}) = \beta\log\pi_{\text{ref}} + r(\mathbf{x},\mathbf{y})$ which is bounded in magnitude as $\beta \to 0$. We also have that if the tempered target and model logprobs are equal, the untempered logprobs must also be equal. Given this, we can now define the tempered loss as follows:

**Definition 4.7.** We define the tempered loss relative to an $f$-divergence $f$ as:

$$\tilde{\mathcal{L}}_{f,\beta}(\Delta_\theta(\mathbf{y})) = \frac{1}{\beta}\mathcal{L}_f(\beta\Delta_\theta(\mathbf{y})) \qquad (7)$$

where $\Delta_\theta(\mathbf{y}) = \log\pi_\theta(\mathbf{y}) - \mathcal{R}_\star(\mathbf{y}|\mathbf{x})$.

The scaling by $\frac{1}{\beta}$ is to ensure the gradient size doesn't vary with $\beta$. The tempered loss formulation also provides a new viewpoint on the KIMI loss (Team et al., 2025a;b) as it is the tempered loss of the reverse $\mathbb{KL}$ loss:

**Example 4.8.** Let $f(u) = u\log u$, then we have the tempered loss is equal to:

$$\tilde{\mathcal{L}}_{f,\beta}(\Delta_\theta(\mathbf{y})) = \frac{1}{2\beta}\mathbb{E}_{\mathbf{y}\sim\pi_\theta}\left[\left(\beta\log\frac{\pi_\theta(\mathbf{y}|\mathbf{x})}{\mathcal{R}(\mathbf{y}|\mathbf{x})} - \log\tilde{Z}\right)^2\right]$$

Which leads to the batch wise normalisation $\widehat{\log Z} = \bar{r}$ used by KIMI under the assumption that $0 \approx \beta\left(\log\pi_\theta\left(\mathbf{y}_i|\mathbf{x}\right) - \log\pi_{\theta_{\text{ref}}}\left(\mathbf{y}_i|\mathbf{x}\right)\right)$, as discussed in 3.

# 5. Generalisation to Generative Flow Networks

We now apply the novel loss family detailed in Section 4 to Generative Flow Networks (GFlowNets) (Bengio et al.,

2021; 2023). GFlowNets are a family of probabilistic methods that amortise sampling from spaces with compositional structure. One of the key benefits of GFlowNets is that they allow for both on and off-policy training, with exploratory benefits from off-policy samples.

In the on-policy case there exist partial equivalences between GFlowNets and hierarchical variational inference (HVI) (Malkin et al., 2023), in the sense that the expected gradients with certain losses (Malkin et al., 2022a) match the expected gradients from performing HVI with the KL. We demonstrate that our losses naturally extend these equivalences to all other $f$-divergences and provide a new family of losses for training GFlowNets on- and off-policy. First, we provide a brief background on GFlowNets, sticking to discrete GFlowNets for simplicity, though GFlowNets have been extended to the continuous case (Lahlou et al., 2023).

## 5.1. Background: GFlowNets, Trajectory Balance, and Variational Inference

Following (Bengio et al., 2021), we define a DAG $\mathcal{G} = (\mathcal{S},\mathcal{A})$ with states $\mathcal{S}$, actions $\mathcal{A}$, initial state $\mathbf{s}_0$, and terminal states $\mathcal{X}$. A complete trajectory $\tau = (\mathbf{s}_0 \to \cdots \to \mathbf{s}_n)$ ends at $\mathbf{x}_\tau \in \mathcal{X}$. GFlowNets aim to sample $\mathbf{x} \in \mathcal{X}$ proportional to a reward $\mathcal{R}(\mathbf{x})$ by learning a forward policy $\pi_F(\cdot|\mathbf{s},\theta)$ from one state to the next. This induces a trajectory distribution $\pi_F(\tau|\theta) = \prod_{(\mathbf{s}_i,\mathbf{s}_{i-1})\in\tau}\pi_F(\mathbf{s}_i|\mathbf{s}_{i-1})$ and a marginal over terminal states $\pi_F(\mathbf{x}_\tau \mid \theta) = \sum_{\tau:\mathbf{x}_\tau=\mathbf{x}}\pi_F(\tau|\theta)$.

Direct likelihood maximization is intractable due to the marginal sum. Malkin et al. (2022a) address this by introducing a learnable $Z$ and backward policy $\pi_B(\tau|\mathbf{x}_\tau,\phi) = \prod\pi_B(\mathbf{s}_{i-1} \mid \mathbf{s}_i,\phi)$ and enforcing the *trajectory balance* constraint:

$$Z\pi_F(\tau|\theta) = \mathcal{R}(\mathbf{x}_\tau)\pi_B(\tau|\mathbf{x}_\tau,\phi).$$

Here $\pi_B(\tau \mid \mathbf{x}_\tau,\phi) = \prod_{(\mathbf{s}_{i-1},\mathbf{s}_i)\in\tau}\pi_B(\mathbf{s}_{i-1} \mid \mathbf{s}_i,\phi)$ and $Z = \sum_{\mathbf{x}\in\mathcal{X}}\mathcal{R}(\mathbf{x})$. Malkin et al. (2022a) show that if $\theta,\phi$ are such that the trajectory balance constraint is satisfied for all $\tau \in \mathcal{T}$, then $\pi_F(\mathbf{x}|\theta) \propto \mathcal{R}(\mathbf{x})$. Defining the log trajectory discrepancy as:

$$\Delta(\tau,\theta,\phi) = \log\frac{Z_\phi\pi_F(\tau|\theta)}{\mathcal{R}(\mathbf{x}_\tau)\pi_B(\tau|\mathbf{x}_\tau,\phi)},$$

where $Z_\phi$ is now a learnable parameter, the trajectory balance constraint is satisfied if $\Delta(\tau,\theta,\phi) = 0$ for all $\tau \in \mathcal{T}$, which leads Malkin et al. (2022a) to define the trajectory balance loss as:

$$\mathcal{L}_{\text{TB}}(\tau,\phi,\theta) := (\Delta(\tau,\theta,\phi))^2. \qquad (8)$$

The parameters, $\theta,\phi$, are then updated using the expected gradients of the loss as $\mathbb{E}_{\mu(\tau)}\left[\nabla\theta\mathcal{L}_{\text{TB}}(\tau,\phi,\theta)\right]$ and

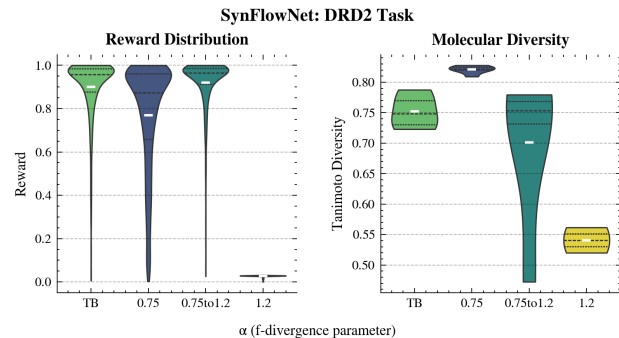

*(a)* Comparison of different losses on the grid task of Bengio et al. (2021). The forward KL and Hellinger are more mode covering than standard trajectory balance, meaning they find all 4 modes and fit the distribution quicker. Pearson is more mode seeking than trajectory balance and attaches to the first mode it finds.

*(b)* Training a SynFlowNet with a variety of losses based on $\alpha$-divergences. In $\alpha$ divergences, $\alpha$ is a controllable parameter with lower values incentivising mode covering and higher values incentivizing mode seeking, standard trajectory balance corresponds to $\alpha = 1$. Annealing $\alpha$ during training allows these to be traded off, lead to the sampling of more diverse high reward molecules.

$\mathbb{E}_{\mu(\tau)}\left[\nabla\phi\mathcal{L}_{\text{TB}}(\tau, \phi, \theta)\right]$, using expected gradients under a behavior policy $\mu(\tau)$, typically a tempered $\pi_F$.

Malkin et al. (2023) relate this to hierarchical variational inference (HVI) (Ranganath et al., 2016), viewing $\pi_B$ as a posterior and $\pi_F$ as a generative model minimizing the $f$-divergence :

$$\mathcal{L}_{\text{HVI,f}}(\pi_F, \pi_B) = \mathcal{D}_f\left(\pi_F\|\pi_B\right) \tag{9}$$

$$= \mathbb{E}_{\tau\sim\pi_B}\left[f\left(\frac{\pi_F(\tau)}{\pi_B(\tau)}\right)\right]. \tag{10}$$

For the reverse KL ($\text{R}-\mathbb{KL}$) and KL, Malkin et al. (2022b) show that there are gradient equivalences between HVI and on-policy GflowNet training as:

$$\nabla_\theta\mathcal{D}_{\text{R}-\mathbb{KL}}(\pi_{B,\phi}\|\pi_{F,\theta}) = \mathbb{E}_{\tau\sim\pi_{F,\theta}}\left[\nabla_\theta\mathcal{L}_{\text{TB}}(\tau, \phi, \theta)\right],$$
$$\nabla_\phi\mathcal{D}_{\mathbb{KL}}(\pi_{B,\phi}\|\pi_{F,\theta}) = \mathbb{E}_{\tau\sim\pi_{B,\phi}}\left[\nabla_\phi\mathcal{L}_{\text{TB}}(\tau, \phi, \theta)\right].$$

If $\mathcal{G}$ is a tree, $\pi_B = 1$, meaning we do not have to learn a backward policy and the gradient equivalence is the same as that discussed in Section 3.

### 5.2. $f$-Trajectory Balance

We now demonstrate that the loss introduced in Section 4 can be applied to enforce the trajectory balance constraints in GFlowNets and that doing so generalises the result of Malkin et al. (2022b) to arbitrary $f$-divergences.

**Proposition 5.1.** *For a convex function, $f$, we have that $\mathcal{L}_f$ applied to log trajectory generalises trajectory balance in the sense that we have:*

$$\nabla_\theta\mathcal{D}_f(\pi_F\|\pi_B) = \mathbb{E}_{\tau\sim\pi_{F,\theta}}\left[\nabla_\theta\mathcal{L}_f(\tau, \phi, \theta)\right]$$
$$\nabla_\phi\mathcal{D}_h(\pi_B\|\pi_F) = \mathbb{E}_{\tau\sim\pi_{B,\phi}}\left[\nabla_\phi\mathcal{L}_f(\tau, \phi, \theta)\right]$$

*where $h$ is defined as:*

$$h(u) = \int_1^u\left(2 - f'\left(\frac{1}{t}\right)\right)dt.$$

We demonstrate in Appendix A.3.3 that the on-policy gradients of the backwards policy do not correspond to $f$-divergence minimisation, as with trajectory balance. In this case their validity comes from the fact that $\mathcal{L}_f$ is a valid loss function. Malkin et al. (2022a) also demonstrate that the trajectory balance loss has lower variance than a REINFORCE estimator as the model approaches the optimum, which we show holds for $f$-trajectory balance in Appendix B.2.

## 6. Experiments

We demonstrate our loss family on four tasks: the synthetic grid task for GFlowNets (Bengio et al., 2021), molecule sampling with SynFlowNet (Cretu et al., 2025), diffusion model tuning (Venkatraman et al., 2024), and asynchronous LLM fine-tuning on GSM8k (Cobbe et al., 2021) and Hendrycks MATH (Hendrycks et al.).

We provide a complete list of $f$-divergences used in Appendix B.2, but most important for understanding our results is the family of $\alpha$-divergences, generated by $f(u) = \frac{u^\alpha - u}{\alpha(\alpha-1)} + \frac{\alpha-1}{\alpha}(u-1)$. The key point about this family is not just that it contains many of the most common $f$-divergences, including the forward and reverse KL when $\alpha$ is 0 and 1 respectively[1], but that the $\alpha$ parameter controls the mode covering vs mode seeking behaviour, with lower values of $\alpha$ leading to more mode covering losses. The $\alpha$ divergence loss, $\mathcal{L}_\alpha$, is given by:

$$\mathcal{L}_\alpha(\Delta) = \frac{1}{(\alpha-1)^2}e^{(\alpha-1)\Delta} - \frac{\Delta}{\alpha-1} - \frac{1}{(\alpha-1)^2}.$$

where trajectory balance emerges in the limit as $\alpha \to 1$

---

[1]The value at these points is defined via the limit.

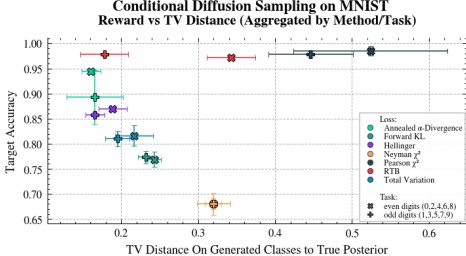

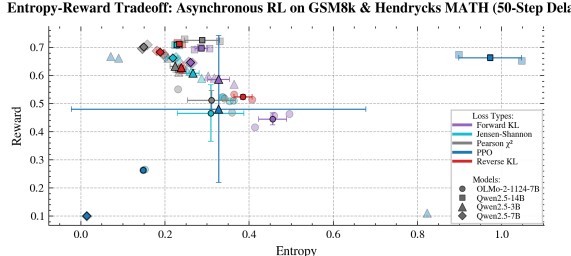

*(a)* Tradeoff in reward vs divergence from the prior when tuning a pretrained diffusion model for MNIST digits to generate odd and even digits respectively.

*(b)* Entropy-Reward Tradeoff when tuning LLMs 50 steps asynchronously with verifiable rewards on GSM8k (Cobbe et al., 2021) and Hendrycks MATH (Hendrycks et al.).

### 6.1. GFlowNet Experiments

#### 6.1.1. SYNTHETIC GRID

We demonstrated this trade-off using the 2D grid experiment from Bengio et al. (2021); Malkin et al. (2022a), where an agent navigates a hypergrid to discover four modes in the grid corners. Actions include moving to adjacent coordinates or terminating the trajectory and low-reward regions between modes simulate the mode-discovery challenges typical of MCMC and sampling algorithms. We evaluated four $\alpha$-divergence losses, $\mathcal{L}_\alpha$, on this task, the forward KL ($\alpha = 0$), Hellinger ($\alpha = 0.5$), reverse KL ($\alpha = 1$), and Pearson $\chi^2$ ($\alpha = 2$). Reverse KL corresponds to training with standard trajectory balance.

Figure 2a demonstrates the mode covering vs mode seeking property, where lower values of $\alpha$ in the forward KL and Hellinger losses lead the agent to find more modes than standard trajectory balance and to find modes quicker. Given one of the key motivations of GFlowNets is its ability to find modes, this clearly demonstrates the benefits of our loss family. On the other hand, the Pearson $\chi^2$ is more mode seeking than trajectory balance, which caused it to get stuck in the first mode it found. This behaviour will in-fact be helpful later for getting reward maximising losses for RL.

#### 6.1.2. SYNFLOWNET

We now apply our losses to tuning SynFlowNets (Cretu et al., 2025), a class of GFlowNets constrained to sample from the space of synthetically accessible molecules. Again the goal is to sample this space proportional to a given reward model, such as predicted binding energy or bioactivity. For these tasks we find that the $\alpha$-divergence losses, $\mathcal{L}_\alpha$, required much less extreme $\alpha$ values for stable training, with $\alpha$ much closer to the trajectory balance value of $\alpha = 1$.

We evaluate $\alpha \in \{0.75, 1.2\}$ against TB across 3 SynFlowNet tasks, sweeping the inverse temperature. $\alpha = 0.75$ yields more diverse molecules than TB while $\alpha = 1.2$ leads to mode collapse. Annealing $\alpha$ from $0.75$ to $1.2$ during training achieves the best of both worlds, as shown for DRD2 in

Figure 2b.

### 6.2. Generative Model Experiments

#### 6.2.1. CONDITIONALLY SAMPLING DIFFUSION MODELS

Whilst we have presented GFlowNets for discrete spaces, they have been extended to continuous ones (Lahlou et al., 2023), and from this applied to tuning of diffusion models (Berner et al.; Venkatraman et al., 2024). Specifically, Venkatraman et al. (2024) view generations as trajectories sampled from a GFlowNet, allowing for a pretrained diffusion model, $p_{\text{prior}}(\mathbf{x})$, to be tuned to an unnormalised posterior distribution $p_{\text{post}}(\mathbf{x}) \propto r(\mathbf{x})p_{\text{prior}}(\mathbf{x})$ without evaluation of the final likelihoods, only the intermediary steps.

To demonstrate that $f$-trajectory balance can be applied in in this case, we repeated the experiments from Venkatraman et al. (2024) by tuning a pre-trained diffusion model for MNIST digits to conditionally sample either odd or even numbers. In this case, the target posterior has multiple modes, one for each target digit, and a correctly fitted posterior would sample them evenly. Figure 3a demonstrates that standard trajectory balance fails to do this, especially for even digits where Appendix B.2 shows it oversamples 0s and 6s. Alternative $f$-trajectory balance losses are able to sample the modes more evenly, with an annealed alpha divergence again getting the best trade-off.

#### 6.2.2. ASYNCHRONOUSLY TUNING LLMS WITH RL

Finally, as a classic example of the entropy-reward trade-off in reinforcement learning with verifiable rewards of large large models, we fine-tune a number of models on a mix of questions on GSM8k (Cobbe et al., 2021) and Hendrycks MATH (Hendrycks et al.). To demonstrate the off policy validity, we train asynchronously with a 50 step delay. For models, we use the Qwen 2.5 family of sizes 3b-14b and OLMo-2-1124-7B, aiming to demonstrate our losses in a variety of model sizes and family's. As we find that standard GRPO cannot be applied 50 steps asynchronously, we compare against the optimised PPO loss implementation from

Intellect (2025), which includes changes such as CISPO clipping (Chen et al., 2025) and DAPO (Yu et al., 2025). For our losses, we use the tempered DevGrad losses (Appendix C) without clipping, importance weighting, or masking.

We train for 300 steps with 3 seeds per configuration; The Reverse KL, Pearson, and Forward KL again demonstrate the same trade-off (Figure 3b). We also include the Jensen-Shannon divergence (requiring numerical normalisation), confirming applicability to the full family of $f$-divergences. The optimised PPO produces inconsistent results across model classes owing to off-policy instability.

## 7. Conclusion

In this work, we have derived a loss family that naturally extends the Vargrad loss for generative models and the trajectory balance loss for GFlowNets by extending their property of on-policy gradients matching the $\mathbb{KL}$ gradients to the whole family of $f$-divergences. We have demonstrated that doing so allows us to control mode-seeking vs mode-covering behaviour in a variety of generative models, whilst training on- and off-policy.

## Acknowledgments

The authors would like to thank Julien Roy and Emmanuel Bengio for feedback on this work.

## Impact Statement

In this work we extended a loss family for an existing model and as so do not foresee any additional concerns beyond the prior work will build on or the impact for a typical probabilistic machine learning paper.

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

# A. Appendix A. Proofs and Derivations

In this appendix, we provide the full derivations for the loss family introduced in Section 4. We prove that the translation invariant loss function $\mathcal{L}_f$ defined in Theorem 4.2 satisfies the two key properties: minimizing the population loss recovers the target distribution (off-policy consistency), and the on-policy auto-differentiated gradients recover the gradients of the associated $f$-divergence. We also establish the inverse mapping described in Theorem 4.4.

## A.1. Proof of Proposition 4.3

**Proposition 4.3.** *Let* $\Delta_\theta(y) = \log p_\theta(y) - \log p_\star(y)$. *The function:*

$$\mathcal{L}_f(\Delta_\theta(y)) = \int_0^{\Delta_\theta(y)} (f'(\exp(t)) - f'(1)) \, dt$$

*is a translation invariant loss function that generalizes the construction in Section 3.2 in that:*

1. *The population loss* $\mathcal{J}_{\mu,f}(\theta) = \mathbb{E}_{y\sim\mu}[\mathcal{L}_f(\Delta_\theta(y))]$ *is minimized when* $p_\theta = p_\star$ *(assuming support coverage).*

2. *If* $\mu = p_\theta$, *the expected auto-differentiated gradients match the* $f$-*divergence gradient:* $\nabla_\theta D_f(p_\theta \| p_\star) = \mathbb{E}_{p_\theta}[\nabla_\theta \mathcal{L}_f(\Delta_\theta(y))]$.

### A.1.1. PROOF OF PART 1: CONVEXITY AND GLOBAL MINIMIZER

Let the scalar loss function with respect to the log-probability difference be denoted by $L(\Delta) = \int_0^\Delta (f'(\exp(t)) - f'(1)) dt$. We determine the properties of $L(\Delta)$ by analyzing its derivatives.

Using the Fundamental Theorem of Calculus, the first derivative is:

$$L'(\Delta) = f'(\exp(\Delta)) - f'(1).$$

Differentiating again with respect to $\Delta$, we obtain the second derivative:

$$L''(\Delta) = f''(\exp(\Delta)) \cdot \exp(\Delta).$$

Since $f$ is a convex function defining an $f$-divergence, $f''(u) \geq 0$ for all $u > 0$. Additionally, the exponential function $\exp(\Delta)$ is strictly positive for all real $\Delta$. Therefore:

$$L''(\Delta) \geq 0 \quad \forall \Delta \in \mathbb{R}.$$

This confirms that $L(\Delta)$ is a convex function. To find the global minimizer, we solve for the stationary point $L'(\Delta) = 0$:

$$f'(\exp(\Delta)) - f'(1) = 0$$
$$f'(\exp(\Delta)) = f'(1).$$

Assuming strict convexity of $f$ (implying $f'$ is strictly monotonic), this equality holds if and only if:

$$\exp(\Delta) = 1 \implies \Delta = 0.$$

Since $L(\Delta)$ is convex and its derivative vanishes at $\Delta = 0$, $\Delta = 0$ is the global minimum. Consequently, the expected population loss $\mathbb{E}_{y\sim\mu}[L(\Delta_\theta(y))]$ is minimized when $\Delta_\theta(y) = 0$ almost everywhere, which implies $\log p_\theta(y) = \log p_\star(y)$, or $p_\theta = p_\star$.

### A.1.2. PROOF OF PART 2: GRADIENT MATCHING CONDITION

We show that the expected gradient of the $f$-divergence on policy equals the expected auto-differentiated gradient of the loss.

**Gradient of the $f$-divergence:** Using the log-derivative trick ($\nabla_\theta p_\theta = p_\theta \nabla_\theta \log p_\theta$) and defining the likelihood ratio $u(y) = p_\theta(y)/p_\star(y)$, the gradient is:

$$\nabla_\theta D_f(p_\theta \| p_\star) = \nabla_\theta \mathbb{E}_{p_\star}[f(u(y))]$$
$$= \mathbb{E}_{p_\star}\left[f'(u(y)) \frac{\nabla_\theta p_\theta(y)}{p_\star(y)}\right] = \mathbb{E}_{p_\theta}[f'(u(y))\nabla_\theta \log p_\theta(y)].$$

Since $\mathbb{E}_{p_\theta}[\nabla_\theta \log p_\theta(y)] = 0$, we can subtract a constant baseline $f'(1)$ without changing the value:

$$\nabla_\theta D_f(p_\theta \| p_\star) = \mathbb{E}_{p_\theta}\left[(f'(u(y)) - f'(1))\nabla_\theta \log p_\theta(y)\right].$$

**Gradient of the Surrogate Loss:** The auto-differentiated gradient of the loss sample $\mathcal{L}_f$ with respect to $\theta$ applies the chain rule to the term inside the integral:

$$\nabla_\theta \mathcal{L}_f(\Delta_\theta(y)) = \frac{\partial \mathcal{L}_f}{\partial \Delta}\nabla_\theta(\log p_\theta(y) - \log p_\star(y))$$
$$= (f'(\exp(\Delta_\theta(y))) - f'(1))\,\nabla_\theta \log p_\theta(y).$$

Substituting $\exp(\Delta_\theta(y)) = u(y)$, we take the expectation over the on-policy samples $y \sim p_\theta$:

$$\mathbb{E}_{p_\theta}[\nabla_\theta \mathcal{L}_f] = \mathbb{E}_{p_\theta}\left[(f'(u(y)) - f'(1))\nabla_\theta \log p_\theta(y)\right].$$

This is identical to the gradient of the $f$-divergence derived above.

### A.2. Proof of Proposition 4.5: Inverse Mapping

**Proposition 4.5.** *Let $\ell : \mathbb{R} \to \mathbb{R}$ be a strictly convex translation invariant loss function. Then the objective $\tilde{\mathcal{J}}_{\mu,\ell}(\theta)$ is minimized when $p_\theta = p_\star$ and corresponds to minimizing an $f$-divergence defined by the generator:*

$$f_\ell(u) = \lambda_1 \int_1^u \ell'(\log t)dt + \lambda_2(u - 1) + c$$

*where constants are chosen to satisfy the standard boundary conditions $f(1) = 0, f'(1) = 1, f''(1) = 1$.*

**Proof.** We seek to find a convex generator $f$ such that the expected on-policy gradients of the divergence $D_f$ are proportional to the expected on-policy gradients of the loss $\ell$.

From the proof of Proposition 4.3, we established that the gradient of an $f$-divergence can be written as an expectation over on-policy samples:

$$\nabla_\theta D_f(p_\theta \| p_\star) = \mathbb{E}_{p_\theta}\left[(f'(u) - f'(1))\nabla_\theta \log p_\theta\right],$$

where $u = p_\theta(y)/p_\star(y) = \exp(y - \hat{y})$.

Now consider the loss $\ell$ acting on the difference in log-probabilities $\Delta = y - \hat{y} = \log u$. The gradient of the loss objective with respect to $\theta$ is:

$$\nabla_\theta \ell(\Delta) = \frac{\partial \ell}{\partial \Delta}\nabla_\theta \Delta$$
$$= \ell'(\log u)\nabla_\theta \log p_\theta.$$

For the gradients to match (up to a scaling factor $\lambda_1$ and an additive constant in the score function which vanishes in expectation), we equate the coefficients of $\nabla_\theta \log p_\theta$:

$$\lambda_1 \ell'(\log u) = f'(u) - C,$$

where $C = f'(1)$. This gives the relationship between the first derivatives. To check convexity, we differentiate with respect to $u$:

$$\lambda_1 \ell''(\log u) \cdot \frac{d}{du}(\log u) = f''(u)$$
$$\lambda_1 \frac{\ell''(\log u)}{u} = f''(u).$$

Since $u > 0$ (probability ratio) and $\ell$ is strictly convex ($\ell'' > 0$), it follows that $f''(u) > 0$ (assuming $\lambda_1 > 0$). Thus, the induced generator $f$ is strictly convex.

To recover the functional form of $f(u)$, we integrate the first derivative relationship $f'(t) = \lambda_1 \ell'(\log t) + C$ from 1 to $u$:

$$f(u) - f(1) = \int_1^u (\lambda_1 \ell'(\log t) + C)\, dt.$$

Imposing the standard $f$-divergence constraint $f(1) = 0$, and letting $\lambda_2 = C$, we obtain:

$$f_\ell(u) = \lambda_1 \int_1^u \ell'(\log t) dt + \lambda_2(u - 1).$$

The constants $\lambda_1$ and $\lambda_2$ can be solved for using the standardization constraints $f'(1) = 1$ and $f''(1) = 1$:

$$f''(1) = 1 \implies \lambda_1 \ell''(0) = 1 \implies \lambda_1 = \frac{1}{\ell''(0)}$$

$$f'(1) = 1 \implies \lambda_1 \ell'(0) + \lambda_2 = 1 \implies \lambda_2 = 1 - \frac{\ell'(0)}{\ell''(0)}.$$

Thus, any strictly convex translation invariant loss on log-probabilities corresponds to a valid, strictly convex $f$-divergence.

### A.2.1. $\mathbb{KL}$ FROM SQUARE LOSS

**Example A.1.** Let $\ell : \mathbb{R} \to \mathbb{R}$ be the squared loss function, so $\ell(x) = x^2$ for $x \in \mathbb{R}$. Then $\ell'(x) = 2x$ and:

$$\begin{aligned}
f_\ell(u) &= \lambda_1 \int_1^u 2\log t\, dt + \lambda_2(u-1) + c \\
&= 2\lambda_1 u \log(u) + \lambda_2(u-1) + c,
\end{aligned}$$

where the boundary conditions give us that $f_\ell(u) = u \log u$.

### A.3. Proof of Proposition 5.1

**Proposition 5.1.** *The following loss:*

$$\mathcal{L}_f(\Delta(\tau, \theta, \phi)) = \int_0^{\Delta(\tau,\theta,\phi)} (f'(\exp(t)) - f'(1))\, dt$$

*generalizes the trajectory balance in the sense that we have:*

$$\nabla_\theta D_f(\pi_F \| \pi_B) = \mathbb{E}_{\tau \sim \pi_{F,\theta}}[\nabla_\theta \mathcal{L}_f(\tau, \phi, \theta)]$$
$$\nabla_\phi D_h(\pi_B \| \pi_F) = \mathbb{E}_{\tau \sim \pi_{B,\phi}}[\nabla_\phi \mathcal{L}_f(\tau, \phi, \theta)]$$

*where $h$ is defined as:*

$$h(u) = \int_1^u \left(2 - f'\left(\frac{1}{t}\right)\right) dt.$$

**Proof.** Let the trajectory log-probability difference be denoted by $\Delta(\tau) = \log \pi_F(\tau|\theta) - \log \pi_B(\tau|\phi)$ (absorbing $Z$ and $R$ into the definition for clarity). Let $u(\tau) = \exp(\Delta(\tau)) = \frac{\pi_F(\tau)}{\pi_B(\tau)}$. The gradient of the scalar loss with respect to $\Delta$ is $\mathcal{L}'(\Delta) = f'(u) - f'(1)$.

### A.3.1. FORWARD POLICY GRADIENT ($\nabla_\theta$)

We examine the expected auto-differentiated gradient of the loss with respect to $\theta$ under the forward policy $\pi_F$:

$$\mathbb{E}_{\tau \sim \pi_F}[\nabla_\theta \mathcal{L}_f] = \mathbb{E}_{\tau \sim \pi_F}\left[\frac{\partial \mathcal{L}_f}{\partial \Delta} \nabla_\theta \Delta(\tau)\right].$$

Since $\pi_B$ does not depend on $\theta$, $\nabla_\theta \Delta(\tau) = \nabla_\theta \log \pi_F(\tau)$. Thus:

$$\mathbb{E}_{\tau \sim \pi_F}[\nabla_\theta \mathcal{L}_f] = \mathbb{E}_{\tau \sim \pi_F}\left[(f'(u) - f'(1))\nabla_\theta \log \pi_F(\tau)\right].$$

From Proposition 4.3, we know that the gradient of the $f$-divergence $D_f(\pi_F \| \pi_B)$ is given by:

$$\nabla_\theta D_f(\pi_F \| \pi_B) = \mathbb{E}_{\tau \sim \pi_F}\left[(f'(u) - c)\nabla_\theta \log \pi_F(\tau)\right],$$

where $c$ is any constant. Choosing $c = f'(1)$ recovers the loss gradient exactly.

A.3.2. BACKWARD POLICY GRADIENT ($\nabla_\phi$)

We examine the expected auto-differentiated gradient of the loss with respect to $\phi$ under the backward policy $\pi_B$:

$$\mathbb{E}_{\tau \sim \pi_B}[\nabla_\phi \mathcal{L}_f] = \mathbb{E}_{\tau \sim \pi_B}\left[\frac{\partial \mathcal{L}_f}{\partial \Delta}\nabla_\phi \Delta(\tau)\right].$$

Since $\pi_F$ does not depend on $\phi$, $\nabla_\phi \Delta(\tau) = -\nabla_\phi \log \pi_B(\tau)$. Thus:

$$\mathbb{E}_{\tau \sim \pi_B}[\nabla_\phi \mathcal{L}_f] = \mathbb{E}_{\tau \sim \pi_B}\left[-(f'(u) - f'(1))\nabla_\phi \log \pi_B(\tau)\right]. \tag{11}$$

Now consider the divergence $D_h(\pi_B \| \pi_F)$ defined by the convex generator $h$. Its gradient with respect to $\phi$ (where $\pi_B$ is the model and $\pi_F$ is the target) is:

$$\nabla_\phi D_h(\pi_B \| \pi_F) = \nabla_\phi \int \pi_F(\tau) h\left(\frac{\pi_B(\tau)}{\pi_F(\tau)}\right) d\tau.$$

Let $v(\tau) = \frac{\pi_B(\tau)}{\pi_F(\tau)} = \frac{1}{u(\tau)}$. Using the derivative rule for $f$-divergences (equivalent to Eq. 54 in the main text but for $h$ and $\pi_B$):

$$\nabla_\phi D_h(\pi_B \| \pi_F) = \mathbb{E}_{\tau \sim \pi_B}\left[(h'(v) - h'(1))\nabla_\phi \log \pi_B(\tau)\right]. \tag{12}$$

To match Eq. (11) and Eq. (12), we require the terms scaling the score function to be equivalent up to a constant shift (which vanishes in expectation). We equate:

$$h'(v) = -f'(u) + C = -f'(1/v) + C.$$

Using the definition of $h$ provided in the proposition:

$$h(v) = \int_1^v \left(2 - f'\left(\frac{1}{t}\right)\right) dt \implies h'(v) = 2 - f'\left(\frac{1}{v}\right).$$

Substituting this into the gradient expectation:

$$\nabla_\phi D_h(\pi_B \| \pi_F) = \mathbb{E}_{\tau \sim \pi_B}\left[(2 - f'(u) - h'(1))\nabla_\phi \log \pi_B(\tau)\right].$$

Absorbing constants 2 and $h'(1)$ into the baseline, this matches the loss gradient term $-(f'(u) - f'(1))$. Thus, the gradients are equivalent.

A.3.3. NON-EXISTENCE OF $f$-DIVERGENCE FOR ON-POLICY BACKWARD GRADIENTS

In Section 5, we established that the gradients for the backward policy $\pi_B$, when sampled from the backward policy correspond to minimizing a valid $f$-divergence $D_h(\pi_B \| \pi_F)$. In this section, we investigate the *on-policy* setting, where samples are drawn from the forward policy $\pi_F$, and we optimize $\pi_B$ to minimize the surrogate loss $\mathcal{L}_f$.

We demonstrate that, unlike the off-policy case, the on-policy gradient update for $\pi_B$ generally **does not** correspond to the descent direction of any valid $f$-divergence. While a generator function $g$ can be derived to match the gradients locally, it fails to satisfy the global convexity requirement ($g'' \geq 0$) for standard choices of $f$, such as the KL divergence.

**A.4. Gradient Matching Setup**

We seek a convex function $g : \mathbb{R}_+ \to \mathbb{R}$ such that the gradient of the divergence $D_g(\pi_F \| \pi_B)$ matches the expected gradient of the surrogate loss $\mathcal{L}_f$ when sampling from $\pi_F$. Note that we define the divergence as $D_g(\pi_F \| \pi_B)$ because the expectation is taken over $\pi_F$.

Let $u(\tau) = \frac{\pi_F(\tau)}{\pi_B(\tau)}$.

**1. Gradient of the Surrogate Loss:** The gradient of the loss $\mathcal{L}_f$ with respect to the backward parameters $\phi$, estimated on-policy, is:

$$\begin{aligned}
\nabla_\phi \mathcal{J}_{\text{on}} &= \mathbb{E}_{\tau \sim \pi_F}\left[\nabla_\phi \mathcal{L}_f(\log u)\right] \\
&= \mathbb{E}_{\tau \sim \pi_F}\left[(f'(u) - f'(1))\nabla_\phi(-\log \pi_B)\right] \\
&= \mathbb{E}_{\tau \sim \pi_F}\left[-f'(u)\nabla_\phi \log \pi_B\right]
\end{aligned}$$

(The constant term $f'(1)$ vanishes because $\mathbb{E}_{\pi_F}[\nabla_\phi \log \pi_B] = 0$).

**2. Gradient of the Candidate Divergence:** Consider the generic divergence $D_g(\pi_F \| \pi_B) = \int \pi_B(\tau) g\left(\frac{\pi_F(\tau)}{\pi_B(\tau)}\right) d\tau$. Differentiating with respect to $\phi$:

$$\nabla_\phi D_g = \int \left(\nabla_\phi \pi_B \cdot g(u) + \pi_B g'(u) \nabla_\phi u\right) d\tau$$

Using the identity $\nabla_\phi u = \pi_F \nabla_\phi(\pi_B^{-1}) = -u \nabla_\phi \log \pi_B$ and $\nabla_\phi \pi_B = \pi_B \nabla_\phi \log \pi_B$:

$$\nabla_\phi D_g = \int \pi_B \left(g(u) - u g'(u)\right) \nabla_\phi \log \pi_B \, d\tau$$

$$= \int \pi_F \frac{1}{u} \left(g(u) - u g'(u)\right) \nabla_\phi \log \pi_B \, d\tau$$

$$= \mathbb{E}_{\tau \sim \pi_F} \left[\left(\frac{g(u)}{u} - g'(u)\right) \nabla_\phi \log \pi_B\right]$$

### A.5. Derivation of the Generator $g$

Equating the terms inside the expectations gives the condition for the gradients to match:

$$-f'(u) = \frac{g(u)}{u} - g'(u)$$

Rearranging into a linear ordinary differential equation for $g(u)$:

$$g'(u) - \frac{1}{u} g(u) = f'(u)$$

Using the integrating factor $I(u) = \exp(\int -\frac{1}{u} du) = \frac{1}{u}$, we solve:

$$\frac{d}{du} \left[\frac{g(u)}{u}\right] = \frac{f'(u)}{u} \tag{13}$$

$$\frac{g(u)}{u} = \int_1^u \frac{f'(t)}{t} dt + C \tag{14}$$

$$g(u) = u \int_1^u \frac{f'(t)}{t} dt + Cu \tag{15}$$

The linear term $Cu$ corresponds to the gradient of a constant expectation and does not affect the convexity analysis.

### A.6. Proof of Non-Convexity

For $D_g$ to be a valid divergence, $g$ must be convex, i.e., $g''(u) \geq 0$ for all $u \in (0, \infty)$. We compute the second derivative of the solution in Eq. 15:

First derivative:

$$g'(u) = \int_1^u \frac{f'(t)}{t} dt + u \left(\frac{f'(u)}{u}\right) = \int_1^u \frac{f'(t)}{t} dt + f'(u)$$

Second derivative:

$$g''(u) = \frac{f'(u)}{u} + f''(u) \tag{16}$$

We now test this condition for the most common case: the Trajectory Balance loss, which corresponds to the KL divergence.

**Counterexample: KL Divergence (Trajectory Balance)** Let $f(u) = u \log u$. Then $f'(u) = 1 + \log u$ and $f''(u) = \frac{1}{u}$. Substituting into Eq. 16:

$$g''(u) = \frac{1 + \log u}{u} + \frac{1}{u}$$

$$g''(u) = \frac{2 + \log u}{u}$$

For $g$ to be convex, we require $g''(u) \geq 0$ for all $u > 0$. However:

$$\frac{2 + \log u}{u} < 0 \iff \log u < -2 \iff u < e^{-2} \approx 0.135$$

Since $u = \pi_F(\tau)/\pi_B(\tau)$, the ratio $u$ can easily fall below $e^{-2}$ in regions where the backward policy assigns significantly higher probability than the forward policy. In this region, $g$ is locally concave.

**Conclusion:** Because $g''(u)$ is not non-negative everywhere, $g$ does not define a valid $f$-divergence. Consequently, the on-policy optimization of the backward policy cannot be interpreted as minimizing a distance measure between distributions in the divergence sense. It is more accurately described as minimizing the variance of the log-ratio or satisfying a moment-matching condition specific to the chosen surrogate loss.

## B. Gradient Variance Analysis

### B.1. $f$-DevGrad Variance Analysis

In this section, we demonstrate that the gradient estimator derived from the batch-wise DevGrad loss, $\mathcal{L}_f^{DG}$, is equivalent to a REINFORCE estimator equipped with a generalized deviation baseline. We prove that the batch normalization step implicitly enforces a zero-sum constraint on the gradient weights, thereby acting as a variance-reducing control variate.

**Gradient Form.** Consider the batch-wise loss over a batch $\mathcal{B} = \{\mathbf{y}_1, \ldots, \mathbf{y}_B\}$ with deviation $\Delta(\mathbf{y}_i) = \log \pi_\theta(\mathbf{y}_i) - \mathcal{R}(\mathbf{y}_i)$:

$$\mathcal{L}_f^{DG}(\mathcal{B}, \theta) = \min_C \frac{1}{B} \sum_{i=1}^B L_f\left(\Delta(\mathbf{y}_i) + C\right).$$

Let $\hat{C}$ be the minimizer of this objective (the batch-wise estimate of $-\log Z$). The gradient of the loss with respect to $\theta$, treating $\hat{C}$ as fixed (via the stop-gradient operator in Eq. 30), is:

$$\hat{\mathbf{g}}^{DG} = \frac{1}{B} \sum_{i=1}^B L_f'\left(\Delta(\mathbf{y}_i) + \hat{C}\right) \nabla_\theta \log \pi_\theta(\mathbf{y}_i).$$

**Zero-Sum Weight Property.** The scalar $\hat{C}$ is defined by the first-order optimality condition of the inner minimization problem:

$$\frac{\partial}{\partial C}\left(\frac{1}{B} \sum_{i=1}^B L_f(\Delta(\mathbf{y}_i) + C)\right)\bigg|_{C=\hat{C}} = \frac{1}{B} \sum_{i=1}^B L_f'(\Delta(\mathbf{y}_i) + \hat{C}) = 0.$$

Let $w_i = L_f'(\Delta(\mathbf{y}_i) + \hat{C})$ be the effective weight for the $i$-th sample. The optimality condition implies $\sum_{i=1}^B w_i = 0$. Consequently, the estimator can be rewritten as a REINFORCE estimator with a sample-dependent baseline:

$$\hat{\mathbf{g}}^{DG} = \frac{1}{B} \sum_{i=1}^B (\tilde{w}_i - \hat{b}) \nabla_\theta \log \pi_\theta(\mathbf{y}_i),$$

where $\tilde{w}_i$ represents the uncentered gradient coefficients derived from $f'$, and $\hat{b}$ is the implicit baseline resulting from $\hat{C}$ such that the coefficients sum to zero.

### B.2. $f$-Trajectory Balance Variance Analysis

We compare the variance of the gradient estimator for the forward policy parameters $\theta$ derived from the $f$-Trajectory Balance loss, $\hat{g}_{f\text{-TB}}$, against the standard score function estimator, $\hat{g}_{\text{SF}}$. Let $\tau$ be a complete trajectory sampled from the forward policy $\pi_F(\cdot|\theta)$. We define the trajectory likelihood ratio as $u(\tau) = \frac{Z_\theta \pi_F(\tau|\theta)}{R(x_\tau)\pi_B(\tau|x_\tau,\phi)}$.

With the score function $S(\tau) = \nabla_\theta \log \pi_F(\tau|\theta)$, the estimators are given by:

$$\hat{g}_{\text{SF}} = f'(u(\tau))S(\tau),$$
$$\hat{g}_{f\text{-TB}} = (f'(u(\tau)) - f'(1))S(\tau) = (f'(u(\tau)) - 1)S(\tau).$$

Since $\mathbb{E}_{\tau \sim \pi_F}[S(\tau)] = 0$, both estimators are unbiased. The difference in their variances is determined by the difference in their expected squared norms:

$$\text{Var}(\hat{g}_{f\text{-TB}}) - \text{Var}(\hat{g}_{\text{SF}}) = \mathbb{E}_{\tau \sim \pi_F}\left[\left((f'(u(\tau)) - 1)^2 - f'(u(\tau))^2\right)\|S(\tau)\|^2\right]$$
$$= \mathbb{E}_{\tau \sim \pi_F}\left[(1 - 2f'(u(\tau)))\|S(\tau)\|^2\right].$$

At the global optimum where the Trajectory Balance constraint is satisfied, we have $Z_\theta \pi_F(\tau) = R(x_\tau)\pi_B(\tau)$, implying $u(\tau) = 1$ for all sampled $\tau$. Given the normalization $f'(1) = 1$, the term inside the expectation becomes:

$$1 - 2f'(1) = -1.$$

Thus, at the optimum, the variance difference is $-\mathbb{E}[\|S(\tau)\|^2]$, which is strictly negative for stochastic policies.

Since $f$ is strictly convex and differentiable, $f'$ is continuous. Consequently, by the continuity of expected values, there exists a neighborhood around the optimum (where $u(\tau) \approx 1$) such that $\mathbb{E}_{\tau \sim \pi_F}[(1 - 2f'(u(\tau)))\|S(\tau)\|^2] < 0$. Therefore, the $f$-Trajectory Balance estimator yields strictly lower variance than the standard score function estimator in the vicinity of the optimum.

## C. Loss Derivations and Properties

In this Appendix, we give the definitions of a variety of standard $f$-divergences and derive the closed form of the $f$-trajectory loss. We also derive the batch wise DevGrad normalisation, the LLM DevGrad formulation, and the tempered LLM DevGrad formulation. We assume the standard normalization $f'(1) = 1$ and $f''(1) = 1$. Reviewing notation we have:

- **Generic:** $\Delta_\theta(\mathbf{y}_i) = \log p_\theta(\mathbf{y}_i) - \mathcal{R}(\mathbf{y}_i)$.

- **LLM Fine-tuning:** The target is the Boltzmann distribution defined by reward $r(\mathbf{x}, \mathbf{y})$ and reference $\pi_{\text{ref}}$.

$$\mathcal{R}(\mathbf{y}_i) = \log \pi_{\text{ref}}(\mathbf{y}_i) + \frac{r(\mathbf{x}, \mathbf{y}_i)}{\beta}$$

The deviation expands to:

$$\Delta_\theta(\mathbf{y}_i) = \log \frac{\pi_\theta(\mathbf{y}_i|\mathbf{x})}{\pi_{\text{ref}}(\mathbf{y}_i|\mathbf{x})} - \frac{r(\mathbf{x}, \mathbf{y}_i)}{\beta}$$

The loss function, $\mathcal{L}_f$, is given by $\mathcal{L}_f(\Delta_\theta(\mathbf{y})) = \int_0^{\Delta_\theta(\mathbf{y})} f'(\exp(t)) - f'(1)dt$. The general Batch-wise DevGrad loss is defined as:

$$\mathcal{L}_f^{\text{DG}}(\mathcal{B}, \theta) = \frac{1}{B}\sum_i \mathcal{L}_f\left(\Delta(\mathbf{y}_i) + \text{SG}\left[\widehat{\log Z}\right]\right)$$

$$\text{where: } \widehat{\log Z} = \arg\min_C \frac{1}{B}\sum_i \mathcal{L}_f\left(\Delta(\mathbf{y}_i) + C\right)$$

$$\text{Which must satisfy: } \frac{1}{B}\sum_i \mathcal{L}'_f\left(\Delta(\mathbf{y}_i) + \widehat{\log Z}\right) = 0$$

Finally, we define the **Stop-Gradient Softmax with Temperature** $\tau$ over the rewards $\mathbf{r}$ as the normalized importance weights:

$$\tilde{\sigma}_\tau(\mathbf{r})_i = \frac{\exp\left(r(\mathbf{x}, \mathbf{y}_i)/\tau\right)}{\text{SG}\left[\sum_{j=1}^B \exp\left(r(\mathbf{x}, \mathbf{y}_j)/\tau\right)\right]}$$

where $\text{SG}[\cdot]$ denotes the stop-gradient operator.

## C.1. Reverse KL Divergence

**Generator:** $f(u) = u \log u$.
**Derivation:** With $f'(u) = 1 + \log u$, the integrand is $t$.

$$\mathcal{L}_{\text{RKL}}(\Delta) = \int_0^\Delta t \, dt = \frac{1}{2}\Delta^2.$$

**Batch-wise Normalization:** The optimality condition $\sum(\Delta_i + \widehat{\log Z}) = 0$ yields the mean difference:

$$\widehat{\log Z} = -\frac{1}{B}\sum_{i=1}^B \Delta(\mathbf{y}_i) \quad \xrightarrow[\text{LLM}]{\pi_\theta \approx \pi_{\text{ref}}} \quad \frac{1}{B}\sum_{i=1}^B \frac{r(\mathbf{x}, \mathbf{y}_i)}{\beta}.$$

**Batch-wise DevGrad Loss:** We substitute the mean difference for the normalization constant.

$$\mathcal{L}_{\text{RKL}}^{\text{DG}}(\mathcal{B}, \theta) = \frac{1}{2B}\sum_{i=1}^B \left( \Delta(\mathbf{y}_i) - \text{SG}\left[ \frac{1}{B}\sum_{j=1}^B \Delta(\mathbf{y}_j) \right] \right)^2.$$

**Batch-wise DevGrad Loss (LLM):** The Vargrad loss on the reward-adjusted log-ratios:

$$\mathcal{L}_{\text{RKL}}^{\text{DG}} = \frac{1}{2}\mathbb{V}\text{ar}\left[ \log\frac{\pi_\theta(\mathbf{y})}{\pi_{\text{ref}}(\mathbf{y})} - \frac{r(\mathbf{x}, \mathbf{y})}{\beta} \right].$$

**Tempered Reverse KL Divergence (KIMI Loss)** From Example 4.10, the tempered loss for the Reverse KL corresponds to the variance of the reward-adjusted log-probabilities scaled by $\beta$. This recovers the standard Kimi setup:

$$\tilde{\mathcal{L}}_{\text{RKL}}^{\text{DG}} = \frac{1}{2\beta}\text{Var}\left[ \beta\log\frac{\pi_\theta(\mathbf{y}|\mathbf{x})}{\pi_{\text{ref}}(\mathbf{y}|\mathbf{x})} - r(\mathbf{x}, \mathbf{y}) \right].$$

## C.2. Forward KL Divergence

**Generator:** $f(u) = 2(u - 1) - \log u$.
**Derivation:** With $f'(u) = 2 - u^{-1}$, the integrand is $1 - e^{-t}$.

$$\mathcal{L}_{\text{FKL}}(\Delta) = \int_0^\Delta (1 - e^{-t}) \, dt = \Delta + e^{-\Delta} - 1.$$

**Batch-wise Normalization:** The condition $\sum(1 - e^{-(\Delta_i + \widehat{\log Z})}) = 0$ implies $\sum e^{-\Delta_i} e^{-\widehat{\log Z}} = B$:

$$\widehat{\log Z} = \log\left( \frac{1}{B}\sum_{i=1}^B e^{-\Delta(\mathbf{y}_i)} \right) \quad \xrightarrow[\text{LLM}]{\pi_\theta \approx \pi_{\text{ref}}} \quad \log\left( \frac{1}{B}\sum_{i=1}^B e^{r(\mathbf{x}, \mathbf{y}_i)/\beta} \right).$$

**Batch-wise DevGrad Loss:** Substituting $\widehat{\log Z}$ into the robust form $\Delta + \widehat{\log Z} + e^{-\Delta}e^{-\widehat{\log Z}} - 1$:

$$\mathcal{L}_{\text{FKL}}^{\text{DG}}(\mathcal{B}, \theta) = \frac{1}{B}\sum_{i=1}^B \left[ \Delta(\mathbf{y}_i) + \text{SG}\left[ \log\left( \frac{1}{B}\sum_{j=1}^B e^{-\Delta(\mathbf{y}_j)} \right) \right] + \frac{e^{-\Delta(\mathbf{y}_i)}}{\text{SG}\left[ \frac{1}{B}\sum_{j=1}^B e^{-\Delta(\mathbf{y}_j)} \right]} - 1 \right].$$

**Batch-wise DevGrad Loss (LLM):** Under the Kimi assumption ($\Delta \approx -r/\beta$), the normalization depends on the rewards with temperature $\beta$.

$$\mathcal{L}_{\text{FKL}}^{\text{DG}} = \frac{1}{B}\sum_{i=1}^B \left[ \log\frac{\pi_\theta(\mathbf{y}_i|\mathbf{x})}{\pi_{\text{ref}}(\mathbf{y}_i|\mathbf{x})(B \cdot \tilde{\sigma}_\beta(\mathbf{r})_i)} + \frac{(B \cdot \tilde{\sigma}_\beta(\mathbf{r})_i)\,\pi_{\text{ref}}(\mathbf{y}_i|\mathbf{x})}{\pi_\theta(\mathbf{y}_i|\mathbf{x})} - 1 \right].$$

**Tempered Forward KL Divergence**  Applying the tempered transformation to Eq. 110. The normalization temperature shifts from $\beta$ to 1.

$$\tilde{\mathcal{L}}_{\text{FKL}}^{\text{DG}} = \frac{1}{B\beta} \sum_{i=1}^{B} \left[ \log \frac{\pi_\theta(\mathbf{y}_i|\mathbf{x})^\beta}{\pi_{\text{ref}}(\mathbf{y}_i|\mathbf{x})^\beta (B \cdot \tilde{\sigma}_1(\mathbf{r})_i)} + (B \cdot \tilde{\sigma}_1(\mathbf{r})_i) \left( \frac{\pi_{\text{ref}}(\mathbf{y}_i|\mathbf{x})}{\pi_\theta(\mathbf{y}_i|\mathbf{x})} \right)^\beta - 1 \right].$$

## C.3. Pearson $\chi^2$ Divergence

**Generator:** $f(u) = \frac{1}{2}(u-1)^2 + (u-1)$.
**Derivation:** With $f'(u) = u$, the integrand is $e^t - 1$.

$$\mathcal{L}_{\chi^2}(\Delta) = \int_0^\Delta (e^t - 1)\, dt = e^\Delta - \Delta - 1.$$

**Batch-wise Normalization:** The condition $\sum(e^{\Delta_i + \widehat{\log Z}} - 1) = 0$ implies $e^{\widehat{\log Z}} \sum e^{\Delta_i} = B$:

$$\widehat{\log Z} = -\log\left( \frac{1}{B} \sum_{i=1}^{B} e^{\Delta(\mathbf{y}_i)} \right) \quad \xrightarrow[\text{LLM}]{\pi_\theta \approx \pi_{\text{ref}}} \quad -\log\left( \frac{1}{B} \sum_{i=1}^{B} e^{-r(\mathbf{x},\mathbf{y}_i)/\beta} \right).$$

**Batch-wise DevGrad Loss:** Substituting $\widehat{\log Z}$ into the robust form $e^\Delta e^{\widehat{\log Z}} - (\Delta + \widehat{\log Z}) - 1$:

$$\mathcal{L}_{\chi^2}^{\text{DG}}(\mathcal{B}, \theta) = \frac{1}{B} \sum_{i=1}^{B} \left[ \frac{e^{\Delta(\mathbf{y}_i)}}{\text{SG}\left[ \frac{1}{B} \sum_{j=1}^{B} e^{\Delta(\mathbf{y}_j)} \right]} - \Delta(\mathbf{y}_i) - \text{SG}\left[ -\log\left( \frac{1}{B} \sum_{j=1}^{B} e^{\Delta(\mathbf{y}_j)} \right) \right] - 1 \right].$$

**Batch-wise DevGrad Loss (LLM):** Using the assumption $\Delta \approx -r/\beta$, the normalization effectively uses a negative temperature $-\beta$.

$$\mathcal{L}_{\chi^2}^{\text{DG}} = \frac{1}{B} \sum_{i=1}^{B} \left[ \frac{\pi_\theta(\mathbf{y}_i|\mathbf{x})}{(B \cdot \tilde{\sigma}_{-\beta}(\mathbf{r})_i)\, \pi_{\text{ref}}(\mathbf{y}_i|\mathbf{x})} - \log \frac{\pi_\theta(\mathbf{y}_i|\mathbf{x})}{(B \cdot \tilde{\sigma}_{-\beta}(\mathbf{r})_i)\, \pi_{\text{ref}}(\mathbf{y}_i|\mathbf{x})} + -1 \right].$$

## C.4. Neyman $\chi^2$ Divergence

**Generator:** $f(u) = \frac{(u-1)^2}{2u} + (u-1)$.
**Properties:** *Mass-covering.* Heavily penalizes under-estimation of the target probability.
**Derivation:** With $f'(u) = \frac{3}{2} - \frac{1}{2}u^{-2}$, the integrand is $\frac{1}{2} - \frac{1}{2}e^{-2t}$.

$$\mathcal{L}_{\text{Ney}}(\Delta) = \frac{1}{2}\Delta + \frac{1}{4}e^{-2\Delta} - \frac{1}{4}.$$

**Batch-wise Normalization:** The condition $\sum(\frac{1}{2} - \frac{1}{2}e^{-2(\Delta_i + \widehat{\log Z})}) = 0$ implies $e^{-2\widehat{\log Z}} \sum e^{-2\Delta_i} = B$:

$$\widehat{\log Z} = \frac{1}{2}\log\left( \frac{1}{B} \sum_{i=1}^{B} e^{-2\Delta(\mathbf{y}_i)} \right) \quad \xrightarrow[\text{LLM}]{\pi_\theta \approx \pi_{\text{ref}}} \quad \frac{1}{2}\log\left( \frac{1}{B} \sum_{i=1}^{B} e^{2r(\mathbf{x},\mathbf{y}_i)/\beta} \right).$$

**Batch-wise DevGrad Loss:** Substituting $\widehat{\log Z}$ into the robust form $\frac{1}{2}(\Delta + \widehat{\log Z}) + \frac{1}{4}e^{-2\Delta}e^{-2\widehat{\log Z}} - \frac{1}{4}$:

$$\mathcal{L}_{\text{Ney}}^{\text{DG}}(\mathcal{B}, \theta) = \frac{1}{B} \sum_{i=1}^{B} \left[ \frac{1}{2}\Delta(\mathbf{y}_i) + \text{SG}\left[ \frac{1}{4}\log\left( \frac{1}{B} \sum_{j=1}^{B} e^{-2\Delta(\mathbf{y}_j)} \right) \right] + \frac{\frac{1}{4}e^{-2\Delta(\mathbf{y}_i)}}{\text{SG}\left[ \frac{1}{B} \sum_{j=1}^{B} e^{-2\Delta(\mathbf{y}_j)} \right]} - \frac{1}{4} \right].$$

**Tempered Neyman $\chi^2$ Divergence**  Applying the tempered transformation to Eq. 120. The normalization temperature shifts from $\beta/2$ to $1/2$.

$$\tilde{\mathcal{L}}_{\text{Ney}}^{\text{DG}} = \frac{1}{B\beta} \sum_{i=1}^{B} \left[ \frac{1}{2}\log \frac{\pi_\theta(\mathbf{y}_i|\mathbf{x})^\beta}{\pi_{\text{ref}}(\mathbf{y}_i|\mathbf{x})^\beta (B \cdot \tilde{\sigma}_{1/2}(\mathbf{r})_i)} + \frac{1}{4}\left( \frac{(B \cdot \tilde{\sigma}_{1/2}(\mathbf{r})_i)\, \pi_{\text{ref}}(\mathbf{y}_i|\mathbf{x})^\beta}{\pi_\theta(\mathbf{y}_i|\mathbf{x})^\beta} \right)^2 - \frac{1}{4} \right].$$

## C.5. Squared Hellinger Distance

**Generator:** $f(u) = 2(\sqrt{u} - 1)^2 + (u - 1)$.

**Properties:** *Mode-covering / Robust.* Balances mode coverage with stability. The gradient is bounded as $\Delta \to -\infty$, unlike Forward KL which grows linearly.

**Derivation:** With $f'(u) = 3 - 2u^{-1/2}$, the integrand is $2 - 2e^{-t/2}$.

$$\mathcal{L}_{\text{H}^2}(\Delta) = \int_0^{\Delta} (2 - 2e^{-t/2})\, dt = 2\Delta + 4e^{-\Delta/2} - 4.$$

**Batch-wise Normalization:** The condition $\sum(2 - 2e^{-(\Delta_i + \widehat{\log Z})/2}) = 0$ implies $e^{-\widehat{\log Z}/2} \sum e^{-\Delta_i/2} = B$:

$$\widehat{\log Z} = 2\log\left(\frac{1}{B}\sum_{i=1}^{B} e^{-\frac{1}{2}\Delta(\mathbf{y}_i)}\right) \xrightarrow[\text{LLM}]{\pi_\theta \approx \pi_{\text{ref}}} 2\log\left(\frac{1}{B}\sum_{i=1}^{B} e^{\frac{1}{2\beta} r(\mathbf{x}, \mathbf{y}_i)}\right).$$

**Batch-wise DevGrad Loss:** Substituting $\widehat{\log Z}$ into the robust form $2(\Delta + \widehat{\log Z}) + 4e^{-\Delta/2} e^{-\widehat{\log Z}/2} - 4$:

$$\mathcal{L}_{\text{H}^2}^{\text{DG}}(\mathcal{B}, \theta) = \frac{1}{B}\sum_{i=1}^{B}\left[2\Delta(\mathbf{y}_i) + \text{SG}\left[4\log\left(\frac{1}{B}\sum_{j=1}^{B} e^{-\frac{1}{2}\Delta(\mathbf{y}_j)}\right)\right] + \frac{4e^{-\frac{1}{2}\Delta(\mathbf{y}_i)}}{\text{SG}\left[\frac{1}{B}\sum_{j=1}^{B} e^{-\frac{1}{2}\Delta(\mathbf{y}_j)}\right]} - 4\right].$$

**Batch-wise DevGrad Loss (LLM):** The normalization term corresponds to a softmax with temperature $2\beta$.

$$\mathcal{L}_{\text{H}^2}^{\text{DG}} = \frac{1}{B}\sum_{i=1}^{B}\left[2\log\frac{\pi_\theta(\mathbf{y}_i|\mathbf{x})}{(B \cdot \tilde{\sigma}_{2\beta}(\mathbf{r})_i)\,\pi_{\text{ref}}(\mathbf{y}_i|\mathbf{x})} + 4\sqrt{(B \cdot \tilde{\sigma}_{2\beta}(\mathbf{r})_i)\frac{\pi_{\text{ref}}(\mathbf{y}_i|\mathbf{x})}{\pi_\theta(\mathbf{y}_i|\mathbf{x})}} - 4\right].$$

**Tempered Squared Hellinger Distance** Applying the tempered transformation to Eq. 117. The normalization temperature shifts from $2\beta$ to $2$.

$$\tilde{\mathcal{L}}_{\text{H}^2}^{\text{DG}} = \frac{1}{B\beta}\sum_{i=1}^{B}\left[2\log\frac{\pi_\theta(\mathbf{y}_i|\mathbf{x})^\beta}{(B \cdot \tilde{\sigma}_2(\mathbf{r})_i)\pi_{\text{ref}}(\mathbf{y}_i|\mathbf{x})^\beta} + 4\sqrt{(B \cdot \tilde{\sigma}_2(\mathbf{r})_i)\left(\frac{\pi_{\text{ref}}(\mathbf{y}_i|\mathbf{x})}{\pi_\theta(\mathbf{y}_i|\mathbf{x})}\right)^\beta} - 4\right].$$

## C.6. Jensen-Shannon Divergence

**Generator:** $f(u) = u\log u - (u+1)\log\frac{u+1}{2} + $ linear terms.

**Derivation:** Using $f'(u) = 2\log(\frac{2u}{u+1}) + 1$, we integrate term-by-term involving the dilogarithm function $\text{Li}_2$.

$$\mathcal{L}_{\text{JSD}}(\Delta) = \Delta^2 + 2\Delta\log 2 + 2\text{Li}_2(-e^\Delta) + \frac{\pi^2}{6}.$$

**Batch-wise Normalization:** The condition $\sum\log(\frac{2\exp(\Delta_i + \widehat{\log Z})}{\exp(\Delta_i + \widehat{\log Z}) + 1}) = 0$ requires finding the root $Z$ of:

$$\prod_{i=1}^{B}\frac{2e^{\Delta_i}Z}{e^{\Delta_i}Z + 1} = 1.$$

This does not have a closed-form solution and requires numerical root-finding.

**Batch-wise DevGrad Loss:** No closed form simplification exists. The loss is computed by numerically solving for $\widehat{\log Z}$ using the root-finding equation above and substituting it back into the loss expression:

$$\mathcal{L}_{\text{JSD}}^{\text{DG}}(\mathcal{B}, \theta) = \frac{1}{B}\sum_{i=1}^{B}\mathcal{L}_{\text{JSD}}(\Delta(\mathbf{y}_i) + \text{SG}\left[\widehat{\log Z}\right]).$$

### C.7. $\alpha$-Divergence

**Generator:** $f(u) = \frac{u^\alpha - u}{\alpha(\alpha-1)} + \frac{\alpha-1}{\alpha}(u-1)$.

**Derivation:** With $f'(u) = \frac{u^{\alpha-1}}{\alpha-1} + \frac{\alpha-2}{\alpha-1}$, the integrand is $\frac{1}{\alpha-1}(e^{(\alpha-1)t} - 1)$.

$$\mathcal{L}_\alpha(\Delta) = \frac{1}{(\alpha-1)^2} e^{(\alpha-1)\Delta} - \frac{\Delta}{\alpha-1} - \frac{1}{(\alpha-1)^2}.$$

**Batch-wise Normalization:** The condition $\sum \left( \frac{e^{(\alpha-1)(\Delta_i + \widehat{\log Z})} - 1}{\alpha-1} \right) = 0$ implies $e^{(\alpha-1)\widehat{\log Z}} \sum e^{(\alpha-1)\Delta_i} = B$. Solving for $\widehat{\log Z}$:

$$\widehat{\log Z} = \frac{1}{1-\alpha} \log \left( \frac{1}{B} \sum_{i=1}^{B} e^{(\alpha-1)\Delta(\mathbf{y}_i)} \right) \quad \xrightarrow[\text{LLM}]{\pi_\theta \approx \pi_{\text{ref}}} \quad \frac{1}{1-\alpha} \log \left( \frac{1}{B} \sum_{i=1}^{B} e^{-(\alpha-1)r(\mathbf{x},\mathbf{y}_i)/\beta} \right).$$

**Batch-wise DevGrad Loss:** Substituting $\widehat{\log Z}$ into the robust form with $\mathcal{C}_\alpha = \frac{1}{(\alpha-1)^2}$:

$$\mathcal{L}_\alpha^{\text{DG}}(\mathcal{B}, \theta) = \frac{1}{B} \sum_{i=1}^{B} \left[ \frac{\mathcal{C}_\alpha e^{(\alpha-1)\Delta(\mathbf{y}_i)}}{\text{SG}\left[ \frac{1}{B} \sum_{j=1}^{B} e^{(\alpha-1)\Delta(\mathbf{y}_j)} \right]} - \frac{\Delta(\mathbf{y}_i) + \text{SG}\left[ \frac{1}{1-\alpha} \log \left( \frac{1}{B} \sum_{j=1}^{B} e^{(\alpha-1)\Delta(\mathbf{y}_j)} \right) \right]}{\alpha-1} - \mathcal{C}_\alpha \right].$$

**Batch-wise DevGrad Loss (LLM):** This effectively normalizes with temperature $\frac{\beta}{1-\alpha}$.

$$\mathcal{L}_\alpha^{\text{DG}} = \frac{1}{B(\alpha-1)^2} \sum_{i=1}^{B} \left[ \left( B \cdot \tilde{\sigma}_{\frac{\beta}{1-\alpha}}(\mathbf{r})_i \right) \left( \frac{\pi_\theta(\mathbf{y}_i|\mathbf{x})}{\pi_{\text{ref}}(\mathbf{y}_i|\mathbf{x})} \right)^{\alpha-1} - (\alpha-1) \log \frac{\pi_\theta(\mathbf{y}_i|\mathbf{x})}{\left( B \cdot \tilde{\sigma}_{\frac{\beta}{1-\alpha}}(\mathbf{r})_i \right) \pi_{\text{ref}}(\mathbf{y}_i|\mathbf{x})} - 1 \right].$$

This unified form shows that $\alpha$ controls the temperature of the reward softmax used for normalization. For Forward KL ($\alpha \to 0$), the temperature is $\beta$; for Pearson ($\alpha = 2$), it is $-\beta$; and for Hellinger ($\alpha = 0.5$), it is $2\beta$.

**General Tempered $\alpha$-Divergence** The general form for any $\alpha$, where the normalization temperature becomes $\frac{1}{1-\alpha}$:

$$\tilde{\mathcal{L}}_\alpha^{\text{DG}} = \frac{1}{B\beta(\alpha-1)^2} \sum_{i=1}^{B} \left[ \frac{C_{\alpha,\text{temp}}}{(B \cdot \tilde{\sigma}_{\frac{1}{1-\alpha}}(\mathbf{r})_i)} \left( \frac{\pi_\theta(\mathbf{y}_i|\mathbf{x})}{\pi_{\text{ref}}(\mathbf{y}_i|\mathbf{x})} \right)^{\beta(\alpha-1)} - (\alpha-1) \log \frac{\pi_\theta(\mathbf{y}_i|\mathbf{x})^\beta}{(B \cdot \tilde{\sigma}_{\frac{1}{1-\alpha}}(\mathbf{r})_i) \pi_{\text{ref}}(\mathbf{y}_i|\mathbf{x})^\beta} - 1 \right].$$

### C.8. Total Variation

**Generator:** $f(u) = |u-1|$.
**Derivation:** Assuming the subgradient convention to satisfy $f'(1) = 1$ and symmetry, we use $f'(u) = \text{sgn}(u-1)$ for $u \neq 1$. The integrand becomes $\text{sgn}(e^t - 1) = \text{sgn}(t)$.

$$\mathcal{L}_{\text{TV}}(\Delta) = \int_0^\Delta \text{sgn}(t) \, dt = |\Delta|.$$

**Batch-wise Normalization:** Minimizing the batch loss corresponds to minimizing the sum of absolute deviations $\sum_i |\Delta(\mathbf{y}_i) + C|$. The minimizer is the median of the inputs with reversed sign:

$$\widehat{\log Z} = -\text{Median}\left( \{\Delta(\mathbf{y}_i)\}_{i=1}^B \right) \quad \xrightarrow[\text{LLM}]{\pi_\theta \approx \pi_{\text{ref}}} \quad \text{Median}\left( \left\{ \frac{r(\mathbf{x},\mathbf{y}_i)}{\beta} \right\}_{i=1}^B \right).$$

**Batch-wise DevGrad Loss:** This results in the Mean Absolute Deviation (MAD) of the log-probability differences:

$$\mathcal{L}_{\text{TV}}^{\text{DG}}(\mathcal{B}, \theta) = \frac{1}{B} \sum_{i=1}^{B} \left| \Delta(\mathbf{y}_i) - \text{SG}\left[ \text{Median}(\{\Delta(\mathbf{y}_j)\}_{j=1}^B) \right] \right|.$$

**Batch-wise DevGrad Loss (LLM):** This results in the Mean Absolute Deviation of the reward-adjusted log-ratios.

$$\mathcal{L}_{\text{TV}}^{\text{DG}} = \frac{1}{B} \sum_{i=1}^{B} \left| \log \frac{\pi_\theta(\mathbf{y}_i|\mathbf{x})}{\pi_{\text{ref}}(\mathbf{y}_i|\mathbf{x})} - \frac{r(\mathbf{x},\mathbf{y}_i)}{\beta} - \text{SG}\left[ \text{Median}\left( \left\{ \log \frac{\pi_\theta(\mathbf{y}_j|\mathbf{x})}{\pi_{\text{ref}}(\mathbf{y}_j|\mathbf{x})} - \frac{r(\mathbf{x},\mathbf{y}_j)}{\beta} \right\}_{j=1}^B \right) \right] \right|.$$

**Tempered Total Variation** Applying the tempered transformation to Eq. 130. The normalization relies on the Median of the tempered deviations $\delta \approx -r$, removing the division by $\beta$ inside the absolute difference.

$$\tilde{\mathcal{L}}_{\text{TV}}^{\text{DG}} = \frac{1}{B\beta} \sum_{i=1}^{B} \left| \beta \log \frac{\pi_\theta(\mathbf{y}_i|\mathbf{x})}{\pi_{\text{ref}}(\mathbf{y}_i|\mathbf{x})} - r(\mathbf{x}, \mathbf{y}_i) - \text{SG}\left[ \text{Median}\left( \left\{ \beta \log \frac{\pi_\theta(\mathbf{y}_j|\mathbf{x})}{\pi_{\text{ref}}(\mathbf{y}_j|\mathbf{x})} - r(\mathbf{x}, \mathbf{y}_j) \right\}_{j=1}^{B} \right) \right] \right|.$$

## D. Gradient Analysis of DevGrad Losses

In this section, we derive the closed-form gradients for the batch-wise DevGrad losses ($\mathcal{L}_f^{DG}$). Recall from Eq. 91 that the gradient of the DevGrad loss with respect to the parameters $\theta$ is given by:

$$\nabla_\theta \mathcal{L}_f^{DG}(\mathcal{B}) = \frac{1}{B} \sum_{i=1}^{B} w_i^{DG} \nabla_\theta \log \pi_\theta(\mathbf{y}_i|\mathbf{x})$$

where the effective gradient weight for the $i$-th sample is $w_i^{DG} = L_f'(\Delta_i + \log Z)$. Due to the optimality condition of $\log Z$ (Eq. 92), these weights always satisfy the zero-sum property $\sum w_i^{DG} = 0$, acting as a control variate.

Below, we define $\delta_i = \Delta(\mathbf{y}_i) + \log Z$ as the normalized deviation. We denote the batch mean operation as $\mathbb{E}_{\mathcal{B}}[\cdot] = \frac{1}{B} \sum_{j=1}^{B}(\cdot)$.

### 1. Reverse KL Divergence (Standard Vargrad)

- **Normalization:** $\log Z = -\mathbb{E}_{\mathcal{B}}[\Delta(\mathbf{y})]$.

- **Gradient Weight:**
$$w_i^{DG} = \Delta_i - \mathbb{E}_{\mathcal{B}}[\Delta(\mathbf{y})]$$

### 2. Forward KL Divergence

- **Normalization:** $\log Z$ satisfies $\mathbb{E}_{\mathcal{B}}[e^{-(\Delta+\log Z)}] = 1 \implies e^{\log Z} = \mathbb{E}_{\mathcal{B}}[e^{-\Delta}]$.

- **Gradient Weight:**
$$w_i^{DG} = 1 - \frac{e^{-\Delta_i}}{\mathbb{E}_{\mathcal{B}}[e^{-\Delta(\mathbf{y})}]}$$

### 3. Pearson $\chi^2$ Divergence

- **Normalization:** $\log Z$ satisfies $\mathbb{E}_{\mathcal{B}}[e^{\Delta+\log Z}] = 1 \implies e^{-\log Z} = \mathbb{E}_{\mathcal{B}}[e^{\Delta}]$.

- **Gradient Weight:**
$$w_i^{DG} = \frac{e^{\Delta_i}}{\mathbb{E}_{\mathcal{B}}[e^{\Delta(\mathbf{y})}]} - 1$$

### 4. Neyman $\chi^2$ Divergence

- **Normalization:** $e^{2\log Z} = \mathbb{E}_{\mathcal{B}}[e^{-2\Delta}]$.

- **Gradient Weight:**
$$w_i^{DG} = \frac{1}{2}\left(1 - \frac{e^{-2\Delta_i}}{\mathbb{E}_{\mathcal{B}}[e^{-2\Delta(\mathbf{y})}]}\right)$$

### 5. Squared Hellinger Distance

- **Normalization:** $e^{\frac{1}{2}\log Z} = \mathbb{E}_{\mathcal{B}}[e^{-\Delta/2}]$.

- **Gradient Weight:**

$$w_i^{DG} = 2\left(1 - \frac{e^{-\Delta_i/2}}{\mathbb{E}_{\mathcal{B}}[e^{-\Delta(\mathbf{y})/2}]}\right)$$

**6. Total Variation**

- **Normalization:** $\log Z = -\text{Median}(\{\Delta_j\})$.

- **Gradient Weight:**

$$w_i^{DG} = \text{sgn}\left(\Delta_i - \text{Median}(\{\Delta(\mathbf{y})\})\right)$$

**7. General $\alpha$-Divergence**

- **Normalization:** $e^{(\alpha-1)\log Z} = (\mathbb{E}_{\mathcal{B}}[e^{(\alpha-1)\Delta}])^{-1}$.

- **Gradient Weight:**

$$w_i^{DG} = \frac{1}{\alpha - 1}\left(\frac{e^{(\alpha-1)\Delta_i}}{\mathbb{E}_{\mathcal{B}}[e^{(\alpha-1)\Delta(\mathbf{y})}]} - 1\right)$$

## E. Minimal Implementation

In this Section. we provide minimal formulations of our loss for both standard cases

```python
import torch, math

def log_z_estimate(delta, name='ReverseKL', alpha=1):
    B = delta.size(0)
    logmeanexp = lambda k: torch.logsumexp(k * delta, 0) - math.log(B)
    return {
        'ReverseKL':      lambda: -delta.mean(),
        'ForwardKL':      lambda: logmeanexp(-1.0),
        'Pearson':        lambda: -logmeanexp(1.0),
        'Hellinger':      lambda: 2.0 * logmeanexp(-0.5),
        'NeymanChi2':     lambda: 0.5 * logmeanexp(-2.0),
        'TotalVariation': lambda: -delta.median(),
        'Alpha':          lambda: (1.0 / (1.0 - alpha)) * logmeanexp(alpha - 1.0)
    }[name]()

def devgrad_loss(delta, name='ReverseKL', alpha=1):
    name = 'ReverseKL' if (name == 'Alpha' and abs(alpha - 1) < 1e-4) else name
    d = delta + log_z_estimate(delta, name, alpha).detach()
    return {
        'ReverseKL':      lambda: 0.5 * d**2,
        'ForwardKL':      lambda: d + (-d).exp() - 1,
        'Pearson':        lambda: d.exp() - d - 1,
        'Hellinger':      lambda: 2*d + 4*(-0.5*d).exp() - 4,
        'NeymanChi2':     lambda: 0.5*d + 0.25*(-2*d).exp() - 0.25,
        'TotalVariation': lambda: d.abs(),
        'Alpha':          lambda: (1/(alpha-1)**2) * ((alpha-1)*d).exp() - d/(alpha-1) -
    (1/(alpha-1)**2)
    }[name]().mean()

def tempered_devgrad_loss(log_pi, log_ref, reward, beta=1.0, name='ReverseKL', alpha=1):
    delta_beta = (beta * (log_pi - log_ref)) - reward
    raw_loss = devgrad_loss(delta_beta, name, alpha)
    return raw_loss / beta
```

*Listing 1.* Implementation of Tempered DevGrad Loss

# F. Experiments

## F.1. Synthetic Grid Experiment

For the synthetic grid experiment, we use same setup as Malkin et al. (2022b), with the following parameters:

- **Dimension** ($D$): The dimensionality of the grid.
$$D = 2$$

- **Side Length** ($H$): The resolution of the grid per dimension.
$$H = 128$$

- **State Space** ($\mathcal{S}$): The set of non-terminating states, representing coordinates in the grid.
$$\mathcal{S}^o = \{0, 1, \ldots, H - 1\}^D$$

  The process begins at the initial state $s_0 = \mathbf{0} = (0, \ldots, 0)$.

- **Exploration Parameter** ($R_0$): A background reward constant that determines the scarcity of the reward signal.
$$R_0 = 0.001$$

- **Reward Function** ($R$): The unnormalized density defined over terminating states $s^\top = (s^1, \ldots, s^D)$. It consists of the background term $R_0$ plus two region-based terms that create modes near the corners.

$$R(s^\top) = R_0 + 0.5 \prod_{d=1}^{D} \mathbb{I}\left[\left|\frac{s^d}{H-1} - 0.5\right| \in (0.25, 0.5]\right] + 2 \prod_{d=1}^{D} \mathbb{I}\left[\left|\frac{s^d}{H-1} - 0.5\right| \in (0.3, 0.4)\right]$$

  where $\mathbb{I}[\cdot]$ is the indicator function which is 1 if the condition holds and 0 otherwise.

- **Action Space**: At any state $s = (s^1, \ldots, s^D)$, the allowed actions are:
  1. Increment a coordinate $d$ by 1: $s \to s + e_d$ (allowed only if $s^d < H - 1$).
  2. Terminate: $s \to s^\top$ (transition to the corresponding terminating state in $\mathcal{X}$).

We also provide the following plots for on policy training with the same losses, showing a similar trend:

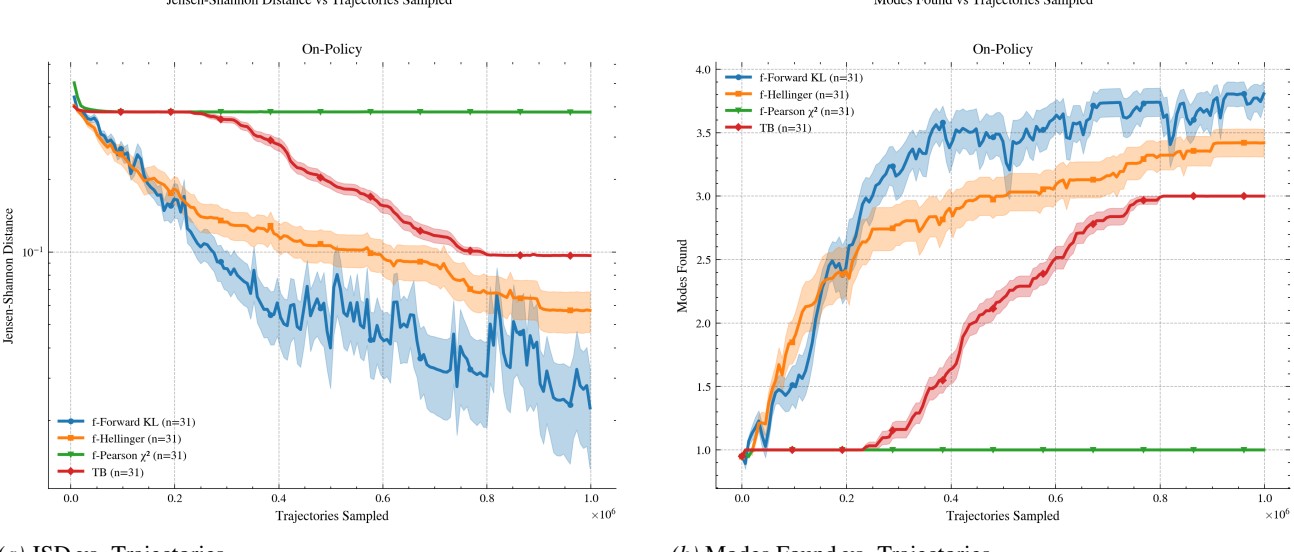

*(a)* JSD vs. Trajectories

*(b)* Modes Found vs. Trajectories

*Figure 4.* An on policy recreation of the synthetic grid experiment in the main text.

## F.2. SynFlowNet Experiments

Here we present our results from the SynFlowNet experiments across three of the tasks from Cretu et al. (2025). These are all using rewards based on bioactivity and binding affinity as:

- sEH (Soluble Epoxide Hydrolase): Uses a pretrained proxy model (a Message Passing Neural Network) trained to predict the AutoDock Vina binding energy.

- GSK3$\beta$ (Glycogen Synthase Kinase-3 Beta): Used as an oracle function from the PMO benchmark to predict bioactivity.

- DRD2 (Dopamine Receptor D2): Used as an oracle function from the PMO benchmark to predict bioactivity.

We present our results across a range of different values of the inverse temperature, $\beta$, in order to understand how our $\alpha$ parameter interacts with the natural way to tune mode-seeking vs covering behaviour in GFlowNets. We fix all the hyperparameter settings used in the original SynFlowNet experiments. Firstly, we present the reward distributions for unique molecules generated and molecule diversity, which indicate the trend that annealing alpha during training ends up with higher reward molecules. This can also be seen from the additional plots of the CDF of the reward function which we add afterward. Finally, we add diversity plots across experimental settings, demonstrating that our losses generally lead to improved diversity.

For each $\alpha, \beta$ setting we train 5 seeds where each seed takes 3 H100 hours to train.

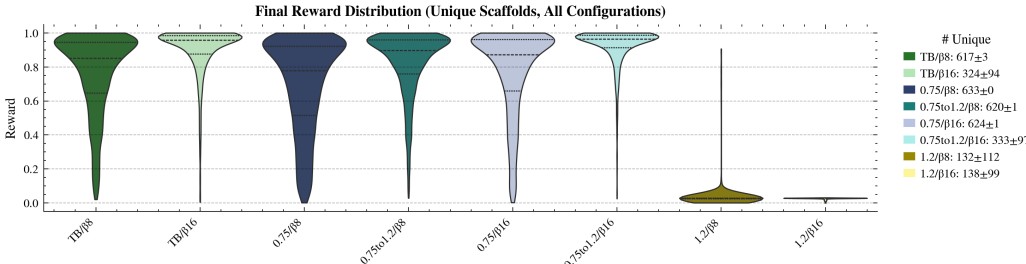

*(a)* DRD2 reward distributions

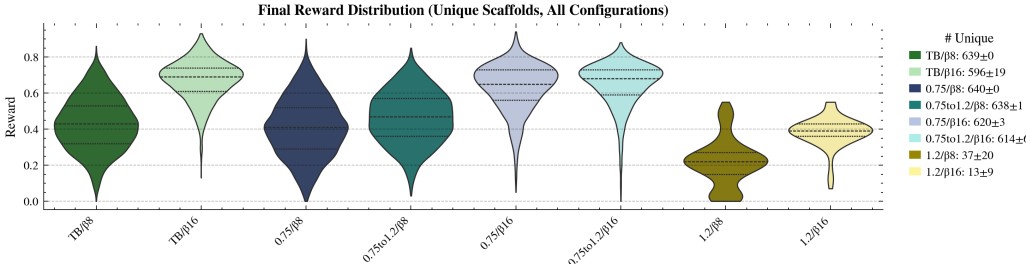

*(b)* GSK3 reward distributions

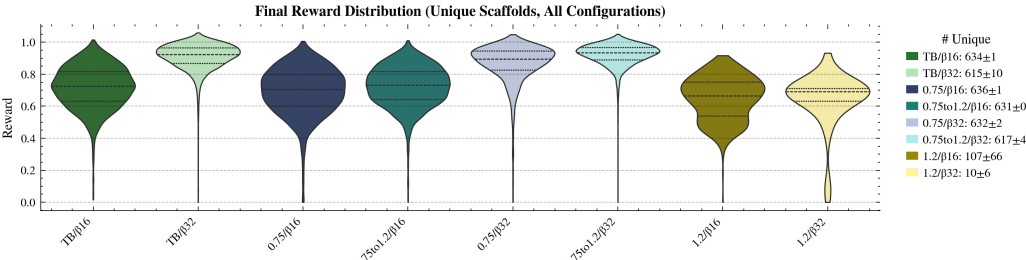

*(c)* sEH reward distributions

*Figure 5.* Swept reward distributions over different alpha beta values in Synflownet training

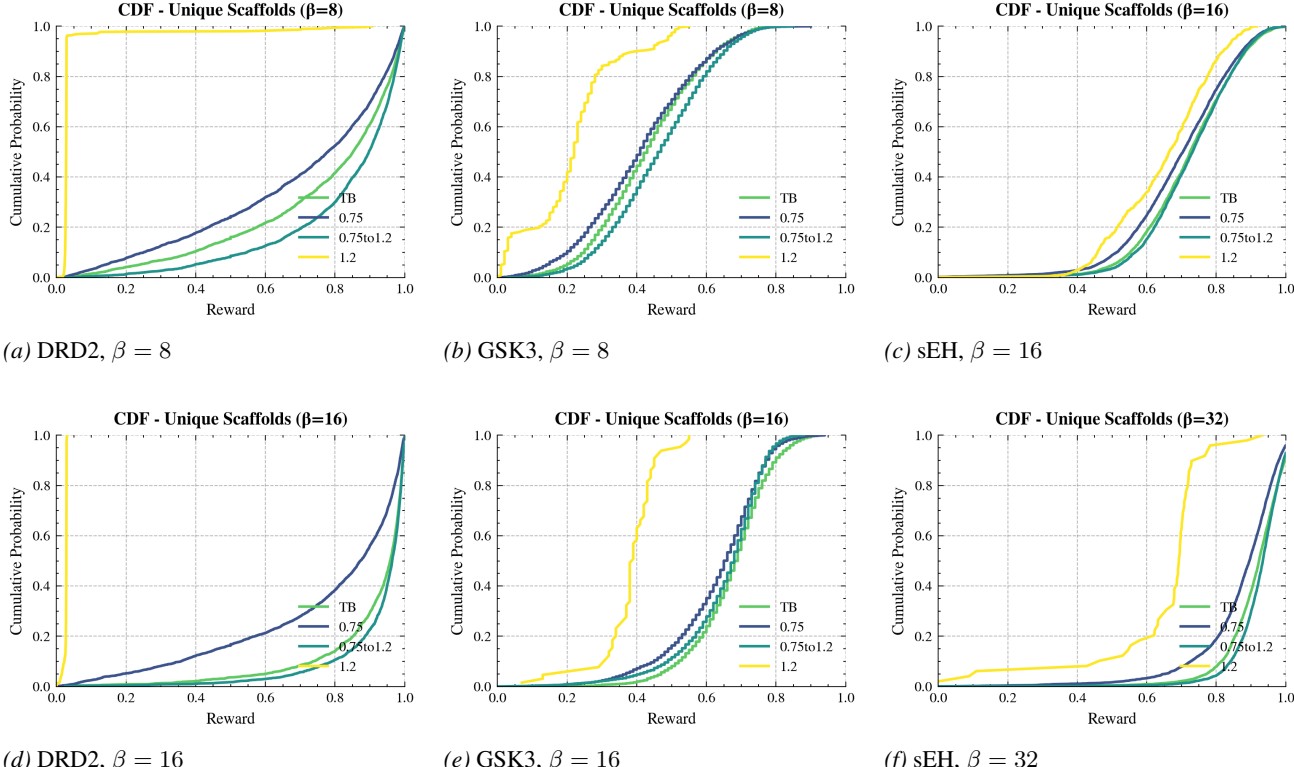

*(a)* DRD2, $\beta = 8$

*(b)* GSK3, $\beta = 8$

*(c)* sEH, $\beta = 16$

*(d)* DRD2, $\beta = 16$

*(e)* GSK3, $\beta = 16$

*(f)* sEH, $\beta = 32$

*Figure 6.* Comparison of scaffold CDFs across different targets (columns) and $\beta$ values (rows) during SynFlowNet training. In 5/6 of the plots the SynFlownet with annealed $\alpha$ has its reward CDF to the right of that trained via trajectory balance, indicating it is generating a higher proportion of high reward molecules. On the other hand $\alpha = 1.2$ leads to clear mode collapse, choosing a few high value molecules.

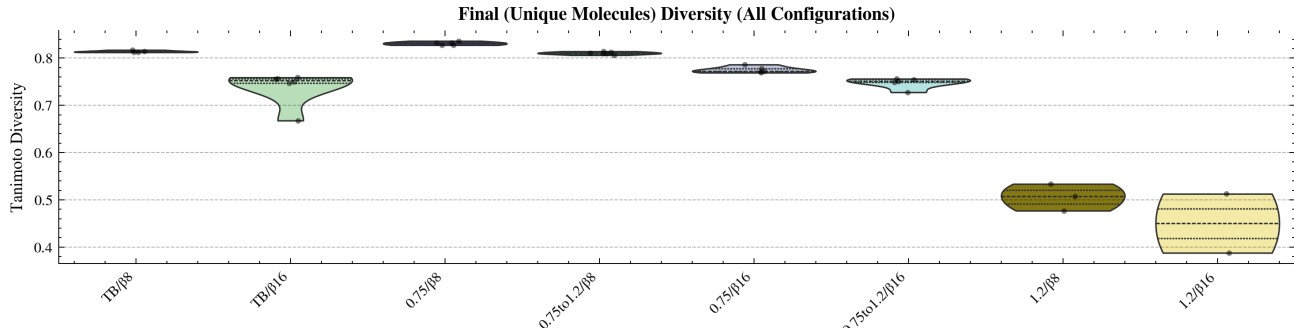

*(a)* GSK3 diversity

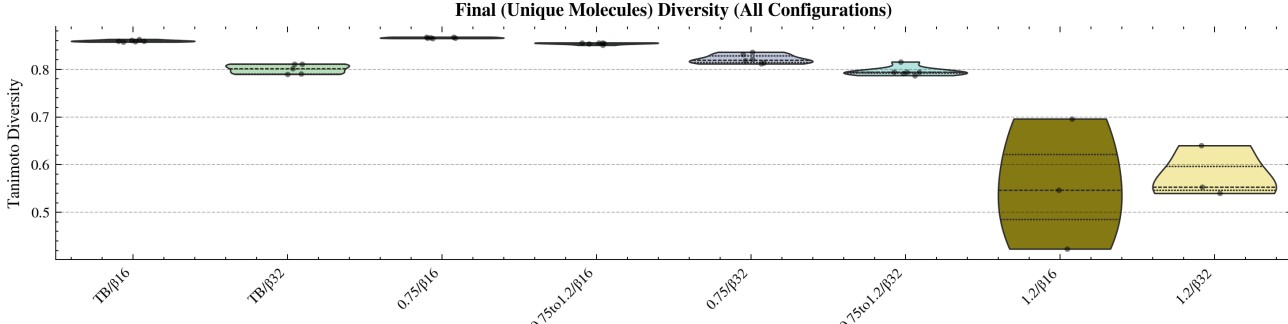

*(b)* sEH diversity

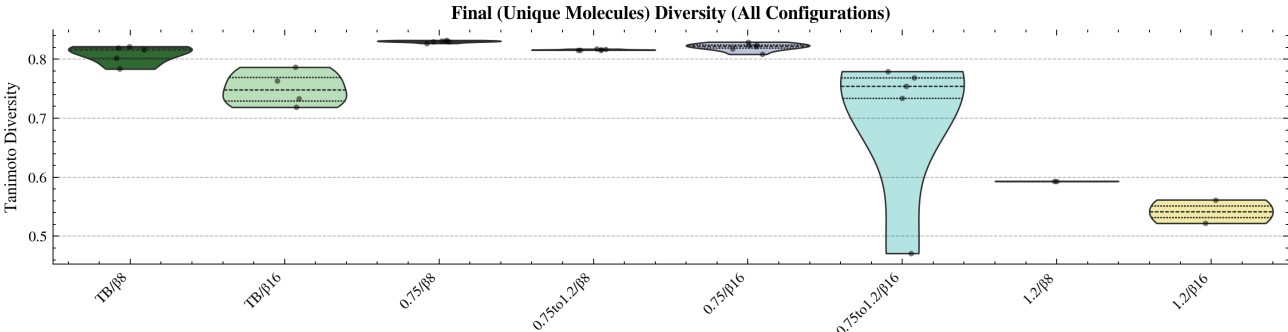

*(c)* DRD2 diversity

*Figure 7.* Tanimoto diversity of unique samples across varying $\alpha$ and $\beta$ values for GSK3, sEH, and DRD2 targets during SynFlowNet training.This demonstrates that lower $\alpha$ leads to more diverse molecules and that annealing can also lead to more diverse molecules across a range of settings.

## F.3. Conditional Sampling in Diffusion Models

Here we present additional details and further analysis for the conditional sampling in diffusion models experiment.

### F.3.1. EXPERIMENTAL DETAILS

Our experimental setup is taken directly from Venkatraman et al. (2024), where the goal is to tune a pre-trained diffusion model for sampling MNIST digits to only sample odd or even digits. This is a helpful task for illustrating the mode seeking vs mode covering properties as to effectively sample the posterior the model must sample from multiple different digits based on their parity. A pre-trained classifier was used to define the reward for generated samples, where the reward is given by:

$$r(\mathbf{x}) = \max_{c \in \text{target\_class}} P(c \mid \mathbf{x}).$$

with $P(c \mid \mathbf{x})$ coming from the pretrained classifier.

**Hyperparameters** We use the same parameters as in Venkatraman et al. (2024) which can be found in the repo at https://github.com/GFNOrg/diffusion-finetuning. For completeness these are:

*Table 1.* Model Training and Optimizer Hyperparameters

| Category | Parameter | Value |
|---|---|---|
| Training | Epochs | 300 |
| | Global Batch Size | 128 |
| | Gradient Accumulation | 1 |
| | Sampling Steps | 200 |
| | Workers | 8 |
| Optimizer (Adam) | Learning Rate ($\eta$) | $6 \times 10^{-4}$ |
| | $\beta_1, \beta_2$ | $0.9, 0.999$ |
| | $\epsilon$ | $1 \times 10^{-8}$ |

### F.3.2. RESULTS ON POSTERIOR FIT

Here we include a variety of results and plots aiming to show how well the different losses capture the posterior. We demonstrate that whilst the relative trajectory balance loss of Venkatraman et al. (2024) leads to the generation of high reward samples (i.e digits with the correct parity) this comes at the cost of sampling unevenly from the posterior digits. This effect can be seen most extremely in the even digits, where relative trajectory balance over samples 0s as they are more distinct from odd digits than other even digits.

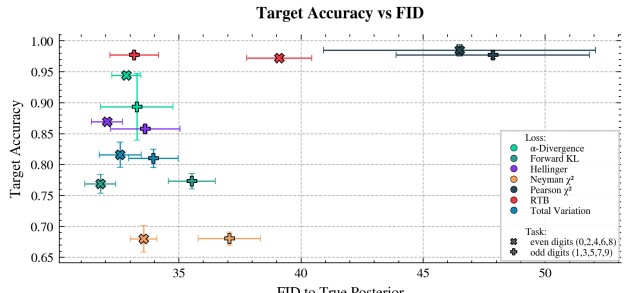

*(a)* Target Accuracy (i.e proportion of target classes sampled) vs FID to posterior by divergence.

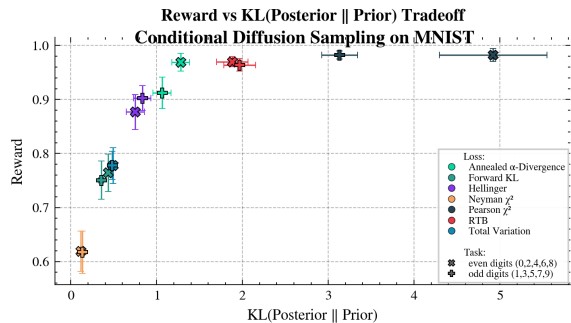

*(b)* Reward (i.e Target accuracy) vs $\mathbb{KL}$ divergence between the fitted posterior and prior over trajectories.

*Figure 8.* Training curves for reward and entropy across all four models. We can see that the large asynchronous delay causes instability in PPO training whereas all *f*-trajectory balance losses lead to stable training .

### F.3.3. GENERATED SAMPLES

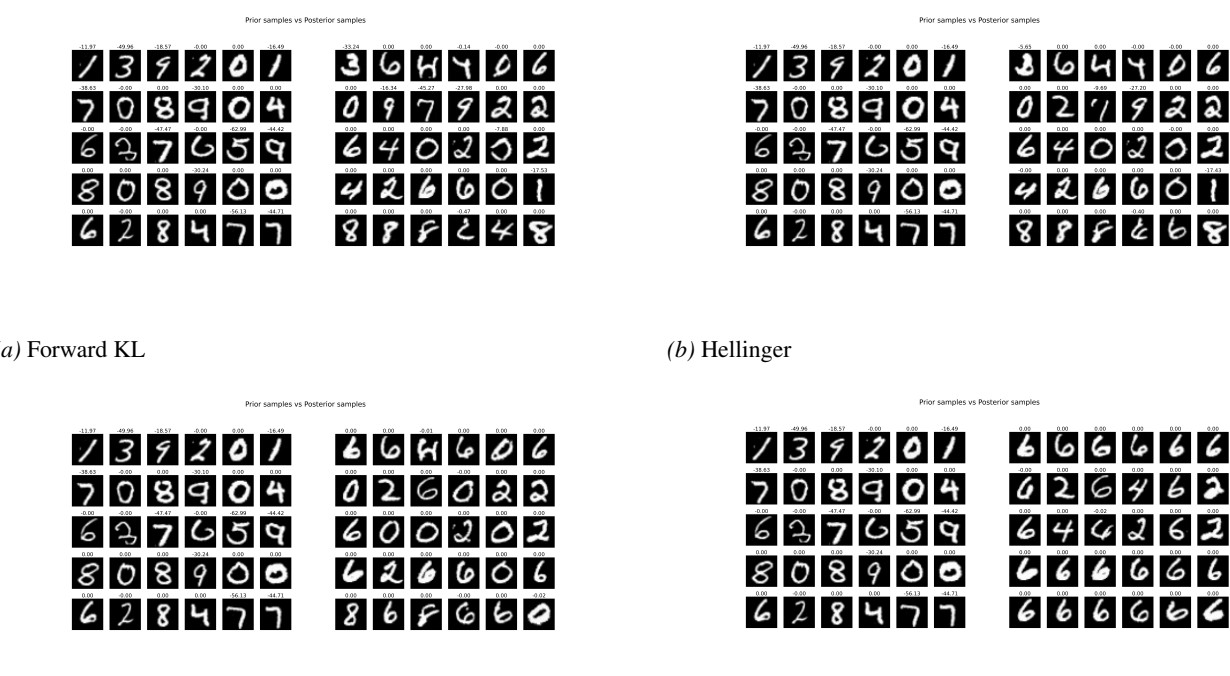

*(a)* Forward KL

*(b)* Hellinger

*(c)* Reverse KL

*(d)* Pearson $\chi^2$

*Figure 9.* Comparison of $f$-divergence losses with more mode covering on the top and mode seeking on the bottom.

Figure 9 shows samples generated by the diffusion model tuned using each loss. We can see that as the reverse KL and Pearson $\chi^2$ are more mode seeking than the Hellinger and Forward KL, we get more mode collapse, leading the model to overly sample 0's or 6's.

### F.4. Asynchronous RLVR

In this section, we provide details and additional plots for the asynchronous RLVR task.

### F.4.1. EXPERIMENTAL DETAILS

To demonstrate our losses, we train on the `math-group` environment from Intellect (2025), which consists of a mix of samples from (Cobbe et al., 2021) and Hendrycks MATH (Hendrycks et al.). For hyperparameters, we use all the defaults from Intellect (2025) with LORA and a 10x multiplier on the learning rate as advocated for in Schulman and Lab (2025). Full hyperparameter details can be found in Table 2.

| Category | Hyperparameter | Value |
|---|---|---|
| General | Max Sequence Length | 4096 |
| | Training Steps | 300 |
| LoRA | Rank ($r$) | 32 |
| | Alpha ($\alpha$) | 64 |
| Optimizer | Optimizer | AdamW |
| | Learning Rate | $1 \times 10^{-5}$ |
| Orchestration | Global Batch Size | 512 |
| | Rollouts per Example | 16 |
| | Async Level | 50 |
| Evaluation | Eval Examples | 600 |
| | Temperature | 1.0 |
| Loss | $\beta$ | 0.001 |
| | Tempered | True |
| | Kimi Approximation | True |

*Table 2.* Model Configuration and Training Hyperparameters. For all hyperparameters not selected we use prime-rl (Intellect, 2025) defaults.

We repeat this task for 4 LLMs, Qwen2.5-3b, -7B, -14B, and OLMo-2-1124-7B with 3 seeds per model. Each model is run on 4 H100s with two for inference and two for training, using prime-rl (Intellect, 2025). Training time varies from 3-5 hours depending on model size. For a comparison loss we use the PPO implementation from prime-rl with all standard hyperparameters, specifically no KL regularisation.

### F.4.2. TRAINING CURVES

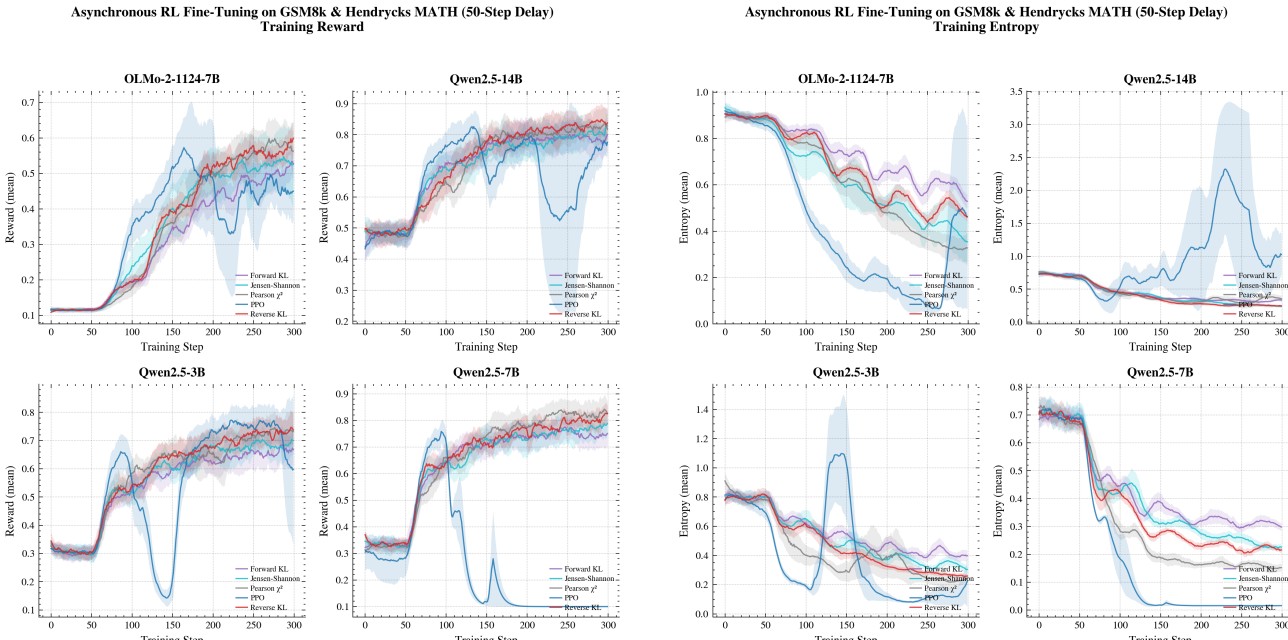

*(a)* Reward training curves averaged over 3 runs, using 10 steps exponential moving average.

*(b)* Entropy training curves averaged over 3 runs, using 10 steps exponential moving average.

*Figure 10.* Training curves for reward and entropy across all four models. We can see that the large asynchronous delay causes instability in PPO training whereas all *f*-trajectory balance losses lead to stable training .

### F.4.3. ENTROPY ON DOWNSTREAM TASKS

We now show that the entropy tradeoff transfers to additional tasks that were not trained on. Specifically on American Invitational Mathematics Examination (AIME) 2024 and OpenPubMedQA Jin et al. (2019).

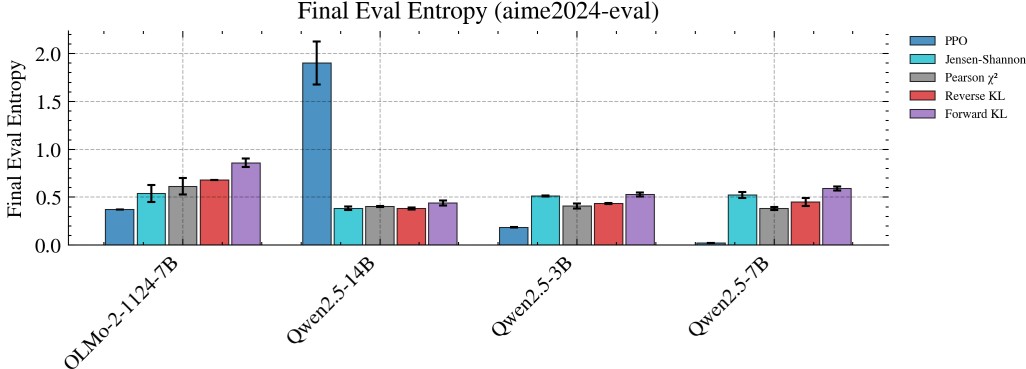

*(a)* Entropy by loss and model on AIME 2024.

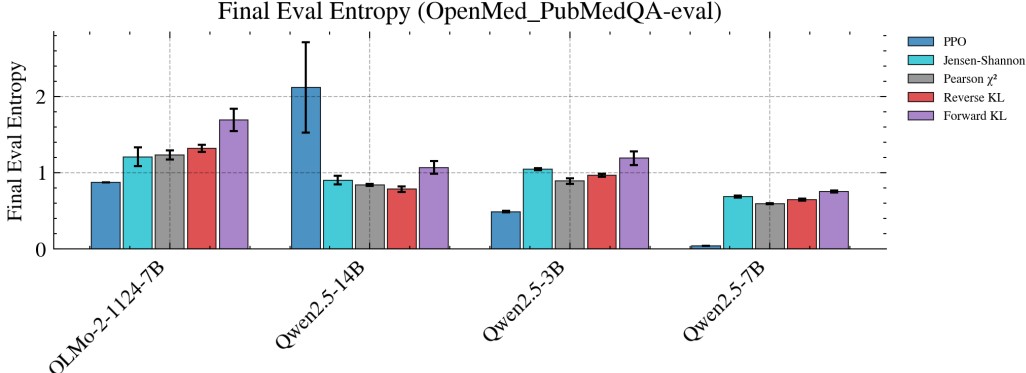

*(b)* Entropy by loss and model on PubMedQA (Jin et al., 2019).

For completeness we include the reward for the fully trained models, however all trained models score very poorly on these tasks.

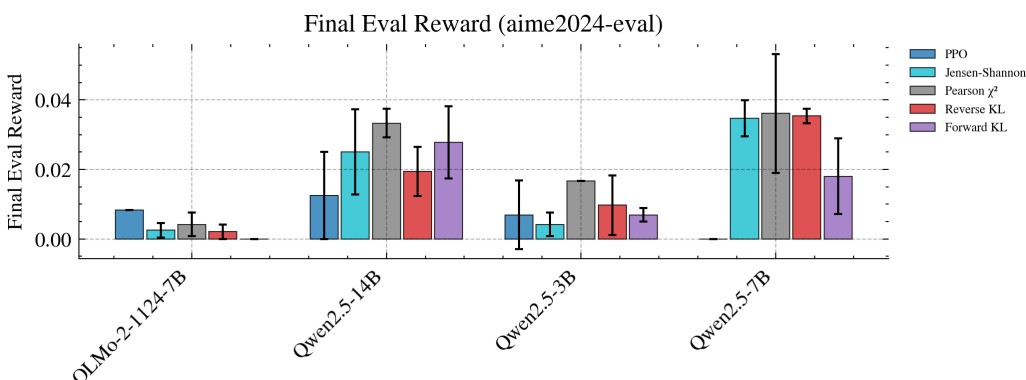

*(a)* Final reward by loss and model on AIME 2024.

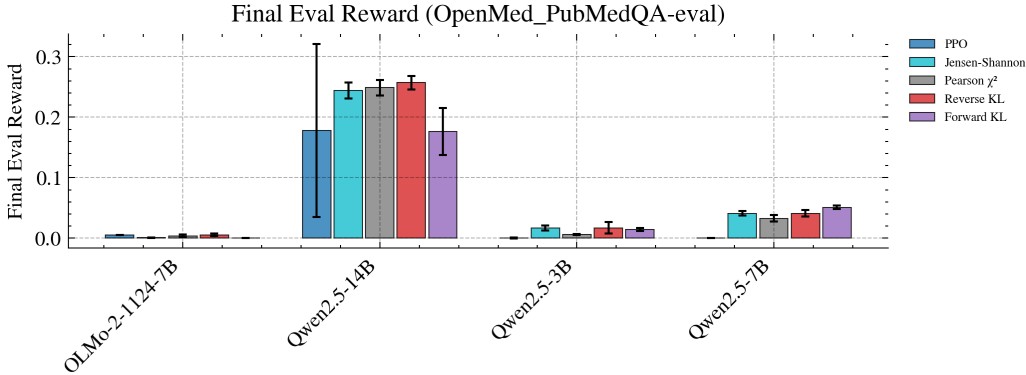

*(b)* Final reward by loss and model on PubMedQA (Jin et al., 2019).

