# OpenReview forum: "$f$-Trajectory Balance: A Loss Family for Tuning GFlowNets, Generative Models, and LLMs with Off- and On-Policy Data"
_ICML.cc/2026/Conference — ICML 2026 regular_

### Official Review · Reviewer_U9s9 · 2026-03-06

**Soundness:** 4
**Presentation:** 3
**Significance:** 3
**Originality:** 3
**Overall Recommendation:** 5
**Confidence:** 2

**Summary:**

The central focus of this paper is the $K2$ estimator of $KL$ divergence proposed by Schulman (2020), which this paper terms $KL_{sq}$. While it is a biased estimator of the $KL$, its gradients on on-policy data are unbiased to the true KL gradients. Further, as it is built on the MSE, it remains a valid loss function with the same global minimizer when applied to off-policy data. Thus, the $KL_{sq}$ estimator offers stability and off-policy compatibility.

This paper generalizes the effectiveness of the $KL_{sq}$ loss by deriving analogous losses for the entire family of $f$-divergences in the
sense that the gradients are equivalent on-policy, but it is also a valid off-policy loss. Specifically, the authors have derived a loss family that naturally extends the Vargrad loss for generative models and the trajectory balance loss for GFlowNets by extending their property of on-policy gradients matching the KL gradients to the whole family of $f$-divergences. This paper demonstrated that doing so allows us to control mode-seeking vs mode-covering behaviour in a variety of generative models, while training on- and off-policy data.

This paper demonstrates the behaviour and utility of this family of loss functions across multiple tasks: Hypergrid, molecule generation, class-conditional sampling in diffusion models, and asynchronous reinforcement learning on language models.

**Compliance With Llm Reviewing Policy:**

Affirmed.

**Final Justification:**

The authors have promised to address my concerns regarding the extensive review of the background literature in their revised manuscript. I had already accepted the paper. Given my low confidence, I won't be able to increase my score further.

**Key Questions For Authors:**

Try to address my comments on the paper's presentation, if possible.

**Limitations:**

I might have missed it, but I could not find any discussion of the limitations of the proposed idea or its potential negative societal impact. Although I am unable to foresee any negative societal impact of this work, I suspect there may be limitations that the authors need to discuss. For example, if the proposed method encounters computational/memory budget issues relative to other approaches when the state space size or trajectory length blow up, etc.

**Strengths And Weaknesses:**

## Soundness
- The paper appears sound to me. All claims are supported by proofs (given in the appendix). Experiments are conducted on multiple tasks to demonstrate the main points.

## Presentation
- By and large, the presentation is good. However, there is some room for improvement. Specifically, it can be written in a way that makes it more accessible to those who are not deeply knowledgeable in areas such as GFlowNet or $f$-divergence. This will help its widespread adoption by the broader AI community. The introduction and contributions can be written more lucidly so that non-experts can follow. Try making fewer assumptions about the audience's heavy mathematical background. This will help the work appeal to the masses.

- Also, there are a few typographical issues, such as a missing equation reference in the paragraph between Equations 28 and 29.

## Significance

- Overall, this work makes a significant contribution to the literature. Extending the known, nicely behaved losses to a broader family while preserving their nice properties certainly gives AI designers much more wiggle room. This opens new avenues for controlling mode-seeking vs. mode-covering behavior during training across a wide range of generative models (on- or off-policy).

## Originality

- While the ideas of $KL_{sq}$ and "minimizing MSE between the log-probabilities of the model and the target" are known, extending loss functions to the $f$-divergence idea seems novel.

---

> ### Author Rebuttal · Authors · 2026-03-30
>
> We thank the reviewer for their positive feedback on the paper. Based on their concerns about the presentation and the heavy mathematical overhead, we will add a separate appendix with a more extensive review of the background literature including $f$-divergences. We will also a a brief discussion that we do not forsee any potential negative societal impact and including some limitations of the method.

---

> > ### Author Rebuttal · Reviewer_U9s9 · 2026-04-03
> >
> > The authors have promised to address my concerns in their revised manuscript. I had already accepted the paper. Given my low confidence, I won't be able to increase my score further.

---

> > > ### Author Response · Authors · 2026-04-08
> > >
> > > We thank the reviewer for their response, for their positive feedback and for noting that all concerns regarding our work have been resolved.

---

### Official Review · Reviewer_64Wt · 2026-03-12

**Soundness:** 3
**Presentation:** 2
**Significance:** 3
**Originality:** 2
**Overall Recommendation:** 4
**Confidence:** 4

**Summary:**

The paper proposes f-Trajectory Balance, a generalization of trajectory balance for Generative Models. The key observation is that the squared loss on log-probability differences used in TB corresponds to minimizing the reverse KL divergence. The authors extend this idea to a family of losses associated with general f-divergences, allowing control over mode-seeking versus mode-covering behavior. The method is evaluated on synthetic tasks, molecule generation, diffusion models, and reinforcement learning fine-tuning of language models.

**Compliance With Llm Reviewing Policy:**

Affirmed.

**Final Justification:**

Given the other reviews and the rebuttal i increased my score. I would however appreciate a more modern evaluations metrics to differentiate between mode-seeking and mass-covering divergence, both on image and text. Such as Precision/Recall/Density/Coverage for the diffusion models or Pass@k for LLMs for instance.

**Key Questions For Authors:**

1.⁠ ⁠Can the authors analyze the variance of the full estimator including the batch estimate of the partition function?
2.⁠ ⁠How does the proposed estimator compare with REINFORCE using standard baselines or advantage estimators?
3.⁠ ⁠Why are precision/recall or coverage-style metrics not used to evaluate mode coverage?
4.⁠ ⁠Can the authors provide larger-scale experiments demonstrating practical benefits?

**Limitations:**

yes

**Strengths And Weaknesses:**

## Strengths

- **Originality**: The paper provides a clean and conceptually appealing connection between translation-invariant losses on log-probability differences and the minimization of general f-divergences. This gives a useful way to reinterpret trajectory balance and extends it beyond its standard reverse-KL view.
- **Significance**: The proposed framework is potentially interesting because it offers a unified way to interpolate between more mode-seeking and more mode-covering objectives, which is a meaningful issue in generative modeling and related sequential decision problems.
- **Soundness**: The main construction is simple and relatively easy to implement, and the paper gives a coherent theoretical motivation for the proposed loss family.

## Weaknesses

- **Soundness**: The variance analysis is incomplete. Appendix B studies the estimator under the assumption that the partition function \(Z\) is known, whereas in practice \(Z\) is estimated from the batch. This introduces additional stochasticity and coupling between samples, so the current analysis does not fully characterize the estimator that is actually optimized during training.
- **Soundness**: Relatedly, the partition-function estimator itself is only weakly justified, and its bias and variance are not analyzed in a satisfactory way.
- **Presentation**: Some parts of the exposition are difficult to follow, and several technical choices would benefit from clearer motivation in the main text. More broadly, the paper would be easier to assess if the discussion of the practical estimator and its limitations were made more explicit.
- **Soundness / empirical validation**: The empirical evaluation remains somewhat limited and relies largely on relatively small-scale or illustrative tasks. This makes it harder to judge the practical importance of the method.
- **Presentation / empirical validation**: Since the paper emphasizes mode-seeking versus mode-covering behavior, it is surprising that the experiments do not include more standard precision/recall or coverage-style metrics for generative models. The current evaluation relies mostly on indirect proxies, which weakens the support for this part of the paper’s claims.
- **Presentation**: The manuscript contains several typographical and editing issues, for example: “that any using any translation invariant” (L.174), “in Section 3 is the from now on the reverse KL and corresponds to” (L.196), “f divergence” instead of “f-divergence” (L.192), “that the its auto-differentiated gradients correspond to the gradients of a corresponding f-divergence” (L.201), “Equation ??” (L.305), and “the the mode covering vs mode seeking behaviour” (L.358).
- **Limitations**: The paper does not sufficiently discuss the limitations induced by the partition-function approximation, the lack of variance analysis for the practical estimator, and the absence of stronger diversity or coverage evaluation metrics.

---

> ### Author Rebuttal · Authors · 2026-03-30
>
> We thank the reviewer for their comments and feedback. To address the reviewers comments:
>
> **Variance Analysis of f-trajectory balance**: In appendix B.2 we analyse the variance under the exact same conditions as [1], assuming that the model is close to convergence but not that the true partition function is known. This is clear from the fact that the estimated partition function, $Z_{\theta}$, is parameterised and lies in direct opposition to the reviewer's claim that “Appendix B studies the estimator under the assumption that the partition function (Z) is known”. Whilst a more general analysis of the variance of this estimator would be of interest to the GFlownet community, to the best of our knowledge it is beyond what is currently known about trajectory balance and as a result beyond the scope of this paper.
>
> **Variance Analysis of DevGrad and relation to REINFORCE**: In Appendix B.1 we demonstrate that the batch wise normalisation implies that the sum of gradient coefficients is zero for each batch. This means that subtracting the mean baseline as in standard REINFORCE approaches for each batch would yield the same estimator, meaning we already get the variance benefits from centring.
>
> **Partition-function estimator itself is only weakly justified**: The justification for the partition function estimator is that if we have reached the optimum, the difference between the target and current logprobs for the generation will be independent of $\mathbf{y}$ and so have deviation zero for any generalised deviation measure. Moreover, we have that if the generalised deviation measure is zero for every batch the target logprobs equal the energy function plus constant, and so taking the exponential the learned distribution is proportional to the the target distribution and therefore equal to it. This demonstrates that we have converged to the correct distribution if and only if the generalised deviation measure is zero for all batches in the support of the sample. We will add this justification more clearly to the draft,
>
> We would also note that we provide extensive experimental justification alongside our theoretical results.
>
> **Presentation**: “ the paper would be easier to assess if the discussion of the practical estimator and its limitations were made more explicit”. We do provide full forms of each estimators practical implementation in Appendix C. More generally if granted acceptance we will use the extra space to improve the text and provide more detail on derivations so they are easier to follow. We have also amended all typos referenced by the reviewer.
>
> **“The empirical evaluation remains somewhat limited and relies largely on relatively small-scale or illustrative tasks.”**  We would like to point the reviewer to the LLM experiments (up to 14 billion parameters, 50 step asyncronous delay) and the Synflownet experiments as clear examples of large scale non toy examples of our framework. Moreover, the illustrative task are present as they clearly demonstrate the properties of our losses on canonical GFlownet tasks, not because these are the only cases where our methods work.
>
> **Metric choices**: The metric choices in this work were motivated by the choice of metrics in the papers in which the experiments were produced. Justifying by experiment:
> 1. Grid task: Here we used the exact same metrics used in [2] as these are standard for this synthetic task.
> 2. Synflownet: In Synflownet training there is no natural notion of precision/recall. Hence we again use the same metrics from the paper which are more optimised for the task at hand and have a more complex notion of diversity based on the Tanimoto distances between molecular fingerprints.
> 3. Diffusion models: Precision/recall would be a helpful addition here, however given we did not log it at the time we cannot rerun the experiments to get it. Having said this, we did track a number of different other metrics during training (FID distance to posterior, total variation distance to posterior on proportion of generated classes, MMD) which we will add to the Appendix.
> 4. Async LLM: For LLMs, given we do not have access to samples from the true posterior we cannot meaningfully get precision recall metrics. Therefore entropy vs reward is commonly used to illustrate such tradeoffs.
>
> Finally, we would comment that we have added significantly greater experimental detail to the Appendix and will release the code to ensure practitioners can reproduce our results with any metrics.
>
> [1] Trajectory balance: Improved credit assignment in GFlowNets. Nikolay Malkin, Moksh Jain, Emmanuel Bengio, Chen Sun, Yoshua Bengio
> [2]- GFlowNets and variational inference. Nikolay Malkin, Salem Lahlou, Tristan Deleu, Xu Ji, Edward Hu, Katie Everett, Dinghuai Zhang, Yoshua Bengio

---

> > ### Author Rebuttal · Reviewer_64Wt · 2026-04-03
> >
> > Questions resolved.

---

> > > ### Author Response · Authors · 2026-04-03
> > >
> > > We thank the reviewer for their response. Given their concerns have been fully resolved we would ask if they would consider raising their score?

---

### Official Review · Reviewer_nUyT · 2026-03-13

**Soundness:** 3
**Presentation:** 3
**Significance:** 2
**Originality:** 3
**Overall Recommendation:** 5
**Confidence:** 4

**Summary:**

The paper proposes a generalization of the squared-log-probability surrogate loss used in trajectory balance and KL-squared objectives for generative model training. The approach consist in the construction valid for the entire family of f-divergences by deriving loss functions over log-probability differences whose on-policy gradients correspond to the gradients of the associated f-divergence while maintaining the same global minimizer when trained off-policy.

The paper presents:
- A theoretical construction mapping  f-divergences to surrogate losses defined over log-probability differences.
- An inverse mapping showing that conditions that certain losses corresponds to minimizing a particular  f-divergence.
- Extensions of Vargrad and trajectory balance used as GFlownet training objectives to arbitrary divergences.
- Applications to multiple domains including: synthetic grid, molecule discovery tasks, diffusion model tuning, and asynchronous reinforcement learning fine-tuning of LLMs.

Empirically, the authors show that choosing different divergences can control the exploration–exploitation trade-off (mode-covering vs mode-seeking) in generative modeling tasks.

**Compliance With Llm Reviewing Policy:**

Affirmed.

**Ethical Review Concerns:**

The previous concern was some prompt injection that appear was not done by the authors.

**Final Justification:**

The author responded my main questions and the main contributions of the paper are clearly delineated as novel compared to existing literature.

**Key Questions For Authors:**

The theoretical development in Sections 4–5 is interesting and well structured. In particular, Propositions 4.1–4.3 establish a correspondence between translation-invariant losses defined on log probability differences and induced $f$-divergence objectives, while Proposition 5.1 connects the resulting loss to the gradient of an $f$-divergence between forward and backward trajectory distributions.

However, several aspects of these results appear closely related to existing loss–divergence correspondences derived in prior work [1,2]. Clarifying the precise relationship between these frameworks would strengthen the theoretical positioning of the paper (and motivate me to update my assessment).

I would appreciate clarification on the following points:

1. Is the forward-gradient identity in Proposition 5.1 derivable from Theorem 4.1 of *Beyond Squared Error* [1] by specializing the weighting over cuts to trajectory balance, i.e. $ w(C) = {1}[C = T] $ and restricting the training objective to the full trajectory as the optimization object? If not, what specific assumption or mathematical step prevents this reduction? In other words, which part of Proposition 5.1 is fundamentally new beyond this specialization?

2. The paper constructs the loss  $ L_f(\Delta) = \int_{0}^{\Delta} (f'(e^t) - f'(1))\,dt $ while [1] derives the inverse mapping using  $ g(t) = f(e^t) - \int_{1}^{e^t} \frac{f(s)}{s}\,ds . $ Are these two constructions intended to represent distinct training objectives, or are they equivalent up to inducing the same on-policy gradient for the forward distribution?

3. How does the loss construction proposed in this paper differ from the divergence-minimization framework used in f-Policy Gradients?

4. Can the gradients derived here be obtained by applying the f-Policy Gradient [2] framework to trajectory distributions rather than state visitation distributions?

5. Does the present formulation offer theoretical or algorithmic advantages beyond those already implied by the general f-divergence policy-gradient [2] framework?

A formal comparison (e.g., showing equivalence or identifying the exact difference in gradient form) would help clarify whether the proposed loss family represents a new class of objectives or a different parameterization of an existing correspondence.

**Limitations:**

yes, the authors adequately address the limitations.

**Strengths And Weaknesses:**

*Soundness*: The paper is generally technically sound. The theoretical development is internally consistent and the derivations appear plausible. In particular, the paper constructs a family of losses

$$
L_f(\Delta) = \int_{0}^{\Delta} (f'(e^t) - f'(1))dt
$$

and shows that the expected gradient of this loss under forward sampling corresponds to the gradient of an $f$-divergence between forward and backward trajectory distributions.  Empirically, the paper evaluates the framework across several domains and the experiments appear competently implemented and illustrate how the choice of divergence affects mode-seeking vs mode-covering behavior.

However, there are two caveats:
- Some theoretical assumptions (e.g., support coverage between policies) are not discussed in detail in the experimental settings.
- The paper does not provide direct comparisons with the most closely related divergence-based training objectives [1,2] (further details in the question to the authors).

---
*Presentation*: The paper is generally well written and the narrative is easy to follow. The structure of the paper  is logical and clear, consisting in first introducing the loss construction, then analyzing its divergence interpretation, and finally applying it to trajectory balance and other settings. The mathematical exposition is reasonably clear, though some derivations (particularly those connecting the loss construction to divergence gradients) could benefit from additional intermediate steps for clarity. The experimental sections are well organized and the figures clearly illustrate the behavioral differences between divergences.

I appreciated in the paper is that it correctly situates its ideas within the emerging line of work on divergence-based training objectives for GFlowNets [3].  This connection is valuable because the current framework can be interpreted as providing a general loss-design perspective for divergence-based training, showing how translation-invariant losses on log probability differences correspond to specific $f$-divergence objectives. However, the positioning relative to prior work could be improved. In particular, closely related results already establish mappings between regression losses on log-ratios and induced $f$-divergence objectives for GFlowNet and policy-gradient training [1,2]. The manuscript would benefit from a clearer discussion of how the current theoretical results differ from these earlier formulations.

Overall, the paper is readable and well structured, but the related work and theoretical discussion should articulate the novelty of the theoretical developments further.

---
*Significance*: the paper addresses the important topic of designing principled loss functions for training GFlowNets. Understanding how different objectives correspond to different divergence measures can provide useful insights into exploration–exploitation trade-offs and the behavior of the training algorithms. The proposed framework may be useful in practice because it provides a systematic way to design losses corresponding to different divergence objectives, rather than relying on ad hoc surrogate losses. However, the impact of the work depends heavily on the degree of novelty in the theoretical framework. If the results are largely a reformulation of existing divergence–loss correspondences, then the contribution may be incremental rather than a major conceptual advance.

---
*Originality*: the main claimed contribution is a general mapping between translation-invariant losses defined on log probability differences and induced $f$-divergence objectives, together with applications to trajectory balance for GFlowNet training. While the formulation is interesting in itself, closely related results already exist in prior work connecting regression losses on log-ratios with induced divergence objectives for GFlowNet training and Policy gradients training in RL [1,2]. In particular, earlier work derives [1] mappings of the form

$$
f(t) = t \int_{1}^{t} \frac{g'(\log s)}{s^2}ds
$$

and

$$
g(t) = f(e^t) - \int_{1}^{e^t} \frac{f(s)}{s}ds,
$$

which already establish a bidirectional correspondence between regression losses (based on $g$) and divergence objectives (for a given $f$).

Another important related work that is currently missing from the discussion is f-Policy Gradients [2]. That work proposes minimizing an $f$-divergence between the agent's state-visitation distribution and a goal distribution and derives the gradient

$$ \nabla_\theta J(\theta) = E_{\tau \sim p_\theta} \left[ \left(\sum_t \nabla_\theta \log \pi_\theta(a_t|s_t)\right) \left(\sum_t f'\left(\frac{p_\theta(s_t)}{p_g(s_t)}\right)\right)\right].
$$

This shows that the gradient of the $f$-divergence objective takes a standard score-function form similar to policy gradients, with weights determined by $f'$.

In the present submission, the authors derive that the gradient of their surrogate loss satisfies

$$ E_{y\sim p_\theta} [\nabla_\theta L_f] = E_{y\sim p_\theta}
[(f'(u)-f'(1)) \nabla_\theta \log p_\theta(y)],
$$

which corresponds to the canonical gradient expression for an $f$-divergence.

Because both works derive policy-gradient-style updates from minimizing $f$-divergence objectives, the relationship between these formulations should be clarified. It is currently unclear whether the theoretical results represent:
- a strict generalization of the earlier framework,
- a reparameterization of an equivalent loss–divergence correspondence,
- or a specialization to trajectory-level objectives.

Clarifying this distinction would strengthen the originality claim.

---
[1] Beyond Squared Error: Exploring Loss Design for Enhanced Training of Generative Flow Networks. ICLR 2025.

[2] f-Policy Gradients: A General Framework for Goal-Conditioned RL using f-Divergences. NeurIPS 2023.

[3] On Divergence Measures for Training GFlowNets. NeurIPS 2024.

---

> ### Author Rebuttal · Authors · 2026-03-30
>
> We would like to thank the reviewer for their comments and for pointing out the very related work, especially [1] which is most closely related and we had missed. We will ensure to add these to the revised version of the paper. To provide a full clarification of the relationships between the works:
>
> The first difference is that our expression works on the ratio of $P_F/P_B$ whereas theirs works with  $P_B/P_F$. Seeing as for any convex $f(t)$ the function $\tilde{f}(t) = 1/t f(1/t)$ is also convex, if we replaced $f(t)$ with $\tilde{f}(t)$ in their expressions we would attain the same result we have for proposition 4.4 and a similar result for the loss function with the expression:
>
> $$ ​​\mathcal{L} _ {f} (\Delta_{\theta}(\mathbf{y}) = ) = \int_0^{\Delta_{\theta} (\mathbf{y}) } {f'(\exp(t))} dt, $$
>
> I.e the same but missing the $f’(1)$ in the integral.
>
> The $f’(1)$ in the integral is important as it is what turns $​​​​\mathcal{L} _ {f} (\Delta_{\theta}(\mathbf{y})$ into a valid loss function, which is what provides the off policy validity guarantees that are crucial for GFlownet training. Without such training, GFlownets are equivalent to hierarchical variational inference. Regarding the properties of $L_f$ (or their equivalent g) all that is commented in [1] is:
>
> Remark 4.2. Note that $f(1)=0, f^{\prime}(1)=g^{\prime}(0)$, and $f^{\prime \prime}(t)=\frac{g^{\prime \prime}(\log t)}{t^2}$. If $g$ is twice differentiable, then $D_f$ is an $f$-divergence if and only if $g$ is convex.
>
> Remark 4.3. Solving for $g$ from $f(t)=t \int_1^t \frac{g^{\prime}(\log s)}{s^2} d s$ and $g(0)=0$ gives $g(t)=f\left(e^t\right)-\int_1^{e^t} \frac{f(s)}{s} d s$.
>
> Therefore there is no proof that the $g$ corresponding to an $f$ is a valid loss function, only the form it satisfies and that it is convex. All other theoretical results in this work relate to on-policy training or proving that there is an f divergence for each loss, not clearly stating that starting with any f provides a loss function as in our result.  In this sense, our work is a generalisation of this, however we did not emphasise off policy training in theorem 5.1 unlike 4.2, which we will change to make more clear. We will also add full references to [1] as the most clearly related work with our differences given as above.
>
> f-policy gradients similarly does not account for off-policy training, with the authors directly commenting that “An avenue for future work could be to develop an off-policy way to solve the objective”. Again our results differ from this by providing proofs of off-policy validity. We note that the derived expression for the gradient of the objective is the gradient of the f-divergence between the distributions $p_\theta$ and $p_g$ (in their notation) which is not something we intend to claim credit for and is a standard fact about f-divergences. We will make this clear as well as the relationship to this paper in the revision.
>
> We would once again like to thank the reviewer for their references, as adding these with the above to our paper will greatly improve the framing. Having said this we would also like to emphasise some contributions that are clearly not included in either of these papers:
>
> 1. The Devgrad estimator, a generalisation of Vargrad and removes the need to learn a normalising constant.
> 2. The tempered loss formulation, a generalisation of the Kimi loss which was crucial when applying our losses to large scale LLM tuning.
> 3. The proofs of the gradients with respect to the parameters of the backward policy in 5.1.
>
> We hope all our the above has helped clarify our results for the reviewer. Please let us know if there are any further questions.

---

> > ### Author Rebuttal · Reviewer_nUyT · 2026-04-01
> >
> > My main question was the connection to prior work, especially one work that was not included in the initial submission. The theoretical connection between the submission and this prior work was well discussed in the rebuttal, with the authors pointing the similarities and dissimilarities, thus showing the unique contribution of this paper and how it expands on a growing theory of generalized losses for GFlowNet training.

---

> > > ### Author Response · Authors · 2026-04-08
> > >
> > > We very much thank the reviewer for their response and are glad that their concerns were addressed by the rebuttal. We would once again say we strongly appreciate the references given by the reviewer and will ensure to add them and full discussion of their relation to our finished work.

---

### Official Review · Reviewer_n6uV · 2026-03-16

**Soundness:** 4
**Presentation:** 2
**Significance:** 4
**Originality:** 3
**Overall Recommendation:** 5
**Confidence:** 4

**Summary:**

This paper proposes a class of surrogate loss functions for unbiased gradient estimation of f-divergence, for both "on-policy" and "off-policy" scenarios (Section 4, before Section 4.1). The key observation is the recent observation made by Tang et al. (2025) that the "squared KL estimate" accidentally gives a correct gradient estimate of the reverse KL divergence when computed with autograd and on-policy samples. This paper provides a strict generalization of the squared loss technique.

The authors consider two applications of this general technique: (1) RLHF and (2) GFlowNets.
- In Section 4.1, the authors then specialize this for the LLM finetuning (RLHF), as a proper generalization of VarGrad for f-divergence minimization, which they call "DevGrad".
- In Section 4.2, they apply the technique for training GFlowNets.
In the experiments, the authors demonstrate that by using the polynomial function as a generator (which corresponds to $\alpha$-divergences), it can trade off mode covering (forward KL) and mode seeking (reverse KL) behaviors.

**Compliance With Llm Reviewing Policy:**

Affirmed.

**Final Justification:**

N/A

**Key Questions For Authors:**

Some minor comments:
- Define $\mathcal{L}_f(\tau,\phi,\theta)$ in Proposition 5.1.
- Please explicitly mention the shorthand "logprobs".
- Please use VarGrad and DevGrad consistently (with capitalization).
- I do not think that $\\mathbb{KL}(\pi_\theta \~\\|\~ \pi\_{\star})(\mathbf{x})$ is a good notation. Please consider using $\\mathbb{KL}(\pi_\theta(\cdot|\mathbf{x}) \~\\|\~ \pi\_{\star}(\cdot|\mathbf{x}))$, or explicitly mention that the notation is used for brevity.

**Limitations:**

This paper does not discuss too much about how to choose what $f$-divergence to use; but this may be beyond the scope of the paper.

**Strengths And Weaknesses:**

## Strengths
While reverse KL divergence is dominant in the literature for distribution matching for obvious reasons, there are benefits of using alternative $f$-divergences. This paper provides a mathematically sound technique to handle this general case. They consider two direct applications of RLHF and GFlowNets.

## Weaknesses
I have some comments about the structure and organization of the paper.
- I think the current flow of the paper is fine, but it could become more easily understandable if Section 3 motivates the problem of estimating the gradient of reverse KL divergence for unconditional distribution $\pi(y)$, and then explain the RLHF application later. The core idea seems to be completely independent from more than half of the content in Section 3, including about the "unnormalized target distribution".
- The title and naming of the paper and methodology is rather weak. $f$-trajectory balance is just one single instance of the more general framework. Moreover, it is only for GFlowNets, meaning that the title is inconsistent to the actual proposal. I believe that the authors should consider a better title.
- I hope that the authors can elaborate the on-policy and off-policy issues more carefully, as this is a very subtle issue that many practitioners might overlook.

---
Overall, I believe that it is a good paper, but I hope the authors can revise the paper to improve the readability.

---

> ### Author Rebuttal · Authors · 2026-03-30
>
> We thank the reviewers for their positive feedback on our work. We have made the changes to notation based on the reviewers suggestions. We would also like to assure the reviewer that we will consider the title and use the extra space if accepted to improve the exposition in Section 3 and elaborate on the on-policy and off-policy issue.

---

> > ### Author Rebuttal · Reviewer_n6uV · 2026-04-04
> >
> > Thanks for the response. Please carefully consider the title and flow to improve the quality of the paper.

---

> > > ### Author Response · Authors · 2026-04-08
> > >
> > > We very much thank the reviewer for their feedback and their positive assessment of our work. We will ensure to take the reviewers points on board regarding flow and title.

---

### Decision · Program_Chairs · 2026-04-30

**Decision:**

Accept (regular)

**Comment:**

This paper generalizes the squared-log-probability surrogate loss to the entire family of f-divergences, enabling control over mode-seeking vs. mode-covering behavior in generative models (RLHF and GFlowNets). The theoretical framework is mathematically sound, and empirical validation covers synthetic tasks, molecule generation, diffusion models, and LLM fine-tuning.

**Strengths:**

- Mathematically sound generalization of f-divergence minimization. (n6uV, nUyT, 64Wt, U9s9)
- Unifies existing methods (VarGrad, Trajectory Balance). (n6uV, 64Wt)
- Diverse empirical validation across multiple tasks. (n6uV, 64Wt, U9s9)

**Weaknesses (resolved in rebuttal):**

- Missing related work (nUyT)
- Incomplete variance analysis / partition function justification (64Wt)
- Limited evaluation metrics (precision/recall for diffusion, Pass@k for LLMs) (64Wt)
- Presentation issues (title, flow, typos) (n6uV, 64Wt, U9s9)

**Additional Comments on Reviewer Discussion:**
The authors have addressed all the concerns from the reviewers.

I suggest accept of the study.